# Characterization of the radiative impact of aerosols on CO₂ and energy fluxes in the Amazon deforestation arch using Artificial Neural Networks

Renato Kerches Braghiere[1,2,3†], Marcia Akemi Yamasoe[3], Nilton Manuel Evora do Rosário[4], Humberto Ribeiro da Rocha[3], José de Souza Nogueira[5], Alessandro C. de Araújo[6,7]

[1]Jet Propulsion Laboratory, California Institute of Technology, 4800 Oak Grove Drive, Pasadena, CA, 91109, USA

[2]Joint Institute for Regional Earth System Science and Engineering, University of California at Los Angeles, Los Angeles, CA, 90095, USA

[3]Departamento de Ciências Atmosféricas, Instituto de Astronomia, Geofísica e Ciências Atmosféricas, Universidade de São Paulo, São Paulo, Brazil

[4]Universidade Federal de São Paulo, Diadema, Brazil

[5]Instituto de Física, Universidade Federal de Mato Grosso, Mato Grosso, Brazil

[6]Embrapa Amazônia Oriental, Travessa Dr. Enéas Pinheiro, s/n, Marco, Caixa Postal 48, Belém, Pará 66095-100, Brazil.

[7]Instituto Nacional de Pesquisas da Amazônia (INPA), Large Scale Biosphere-Atmosphere Experiment in Amazonia (LBA), Avenida André Araújo, 2936, Manaus, Amazonas, 69060-001, Brazil.

*Correspondance to*: Renato K. Braghiere (renato.braghiere@gmail.com)

†Current address: NASA Jet Propulsion Laboratory, M/S 233-305F, 4800 Oak Grove Drive, Pasadena, CA, 91109, USA

**Abstract.** In vegetation canopies with complex architectures, diffuse solar radiation can enhance carbon assimilation through photosynthesis because isotropic light is able to reach deeper layers of the canopy. Although this effect has been studied in the past decade, the mechanisms and impacts of this enhancement over South America remain poorly understood. Over the Amazon deforestation arch large amounts of aerosols are released into the atmosphere due to biomass burning, which provides an ideal scenario for further investigation of this phenomenon in the presence of canopies with complex architecture. In this paper, the relation of aerosol optical depth and surface fluxes of mass and energy are evaluated over three study sites with Artificial Neural Networks and radiative transfer modeling. Results indicate a significant effect of the aerosol on flux of carbon dioxide between the vegetation and the atmosphere, as well as on energy exchange, including that surface fluxes are

sensitive to second order radiative impacts of aerosols on temperature, humidity, and friction velocity. $CO_2$ exchanges increased in the presence of aerosol in up to 55% in sites with complex canopy architecture. A decrease of approximately 12% was observed for a site with shorter vegetation. Energy fluxes were negatively impacted by aerosols over all study sites.

## 1 Introduction

The composition of the atmosphere is determined by the emission and transport of gases and aerosols at different scales, chemical and microphysical processes, wet and dry deposition, or by the distribution of the land surface and oceanic ecosystems around the globe. These processes are represented by the biogeochemical cycle and they involve interactions between different components of the Earth system. These interactions are generally non-

linear and might produce negative or positive radiative forcing signals in the climate system (Forster et al., 2007). The state of the climate system is governed by the energy balance, which is defined as the amount of energy going into the climate system minus the amount of energy dissipated by it mostly through the emission of long wave radiation. In the long term, the amount of solar radiation absorbed by Earth's atmosphere and surface is balanced out by the same amount of longwave radiation emitted by them. About half of the solar radiation is absorbed by

the Earth's surface. This energy is transferred to the atmosphere by heating the air in contact with the surface, through evapotranspiration and longwave radiation emission, which can be absorbed by clouds and greenhouse gases. The atmosphere in turn emits longwave energy back to the surface as well as to the space (Kiehl and Trenberth, 1997).

The Earth's climate system has undergone a number of changing processes considered 'natural' throughout the

history of the planet. The average temperature of the planet has always been directly linked to the amount of $CO_2$ in the atmosphere. The uncertainties on the energy balance associated with greenhouse gases have been reduced over the past years; however, uncertainties associated with atmospheric aerosols remain substantial, despite the various advances achieved since the first studies relating aerosols and the climate system (Boucher et al., 2013). Aerosols are technically defined as solid or liquid particles suspended in the air (Seinfeld and Pandis, 2006), that

can be emitted from natural or anthropogenic sources. They can be found in different shapes and chemical composition, which may vary according to their source of emission and/or processes they are subjected once in the atmosphere. They can also be classified into primary, i.e., when they are directly emitted, or secondary, i.e., when they are formed in the atmosphere through a physical-chemical process known as gas-particle conversion. The typical diameter of aerosols ranges from a few nanometers to tens of micrometers (Seinfeld and Pandis, 2006)

and it can impact their time spent in the atmosphere and their ability to interact with radiation.

Aerosols can travel long distances which increase the time they are exposed to radiation and the probability of interaction with clouds. In addition, the accumulation mode (0.1 μm < particle size < 2.5 μm) interacts most effectively with solar radiation, although the nuclei-mode aerosols (0.1 μm > particle size) are mostly found over the continents and coastal areas, directly related to human activities. Nuclei-mode aerosols account for about 10% of the aerosols of the globe (Textor et al., 2006). Aerosols emitted by natural sources are the dominant type in the climate system. In climate models, aerosols are generally divided into five general categories: dust, marine spray, sulphur aerosols or sulphate, and carbon compounds, which are divided into two subcategories: particulate organic material and Black Carbon (Chin et al., 2002).

The direct radiative effect of aerosols can absorb and scatter radiation in the shortwave band and longwave band, which directly impacts the radiative balance of the climate system (Forster et al., 2007). The net impact of the direct radiative effect of aerosols decreases the available solar energy at the surface. In the last decades, there has been a 2.7% reduction observed on the incident direct solar irradiance at the surface, a phenomenon known as 'Global Dimming' (Stanhill and Cohen, 2001). Although the mechanisms associated with the direct radiative effect of aerosols are well described in the literature, uncertainties are still significant, especially with respect to the spatial and temporal distribution of the aerosols in the atmosphere (Forster et al., 2007; Fletcher et al., 2018). In addition, the radiative effects of aerosols can impact different compartments of the climate system, such as the biosphere, for example.

The biosphere interacts with the climate system through biogeophysical and biogeochemical processes, which can cause both, positive and negative responses on the radiative balance, and some of these responses, especially at a local scale, can be quite significant (Wang et al., 2014; He et al., 2017; Persad and Caldeira, 2018).

The overall reduction of incident radiation at the surface has negatively impacted gross primary productivity (GPP) in light limited ecosystems (Black et al., 2006; Unger et al., 2017; Ezhova et al., 2018). In addition, a decrease in total solar radiation is followed by an increase in its diffuse component, which in turn can increase plant productivity due to a higher production efficiency per unit of incident radiation in the canopy, an effect known as 'the diffuse fertilization effect' (Kanniah et al., 2012). In ecosystems with densely vegetated canopies, the direct shortwave radiation reaches the upper part of the vegetation canopy, while most leaves at medium canopy height or at bottom canopy layers remain shaded. Carbon assimilation increases under atmospheric conditions with moderate aerosol load and high fraction of diffuse radiation (Niyogi et al., 2004) and decreases as the concentration of aerosols increases to a level where the overall radiation is substantially reduced (Yamasoe et al., 2006).

The spatial distribution and nature of the shortwave radiation is critical for the productivity of terrestrial ecosystems and further studies on the impacts of light limitation on vegetation productivity are needed. In light limited ecosystems such as rainforests and temperate forests, the radiative impact of aerosols on vegetation productivity must be evaluated in combination with other environmental factors, such as temperature and water (O'Sullivan et al., 2016.). With fieldwork and numerical studies in the Australian savannah, Whitley et al. (2011) suggested that vegetation productivity was more influenced by light limitation than by water limitation. Actually, the response of photosynthetic capacity to an increase in the concentration of atmospheric aerosol has been extensively studied over almost the past two decades (Roderick et al., 2001; Gu et al., 2002; Niyogi et al., 2004; Yamasoe et al., 2006; Oliveira et al., 2007; Mercado et al., 2009; Doughty et al., 2010; Kanniah et al., 2010; Zhang et al., 2010; Kanniah et al., 2012; Li et al., 2014; Rap et al., 2015; Tang and Dubayah, 2017) through observations (Steiner et al., 2013) and modeling (Moreira et al., 2017) frameworks.

In this paper, however, we investigate the impact of atmospheric aerosols on the carbon balance over three sites in South America, in the deforestation arch of the Amazon forest, making use of a novel approach, combining two different types of modeling techniques. First, we make use of Artificial Neural Networks (ANNs), a machine learning algorithm that can approximate any nonlinear deterministic function (Gentine et al., 2018),  to build a new modeling framework from scratch relating a number of state variables (radiation, temperature, humidity, as well as the aerosol optical depth) with energy and mass fluxes for three study sites representing a range of structurally different ecosystems in the Amazon; and second, a radiative transfer scheme driven with a unique aerosol optical model developed in Rosário et al. (2009) and described in Rosário et al. (2011). A recent study has performed a modeling evaluation of the radiative impact of aerosols over a temperate forest with complex structure in the USA (Lee et al., 2018), but the studies in the Amazon remain mostly observational.

Machine learning has significantly advanced in the last decades, being applied to a number of different research areas, and more recently used in Earth System studies as a powerful tool to link physical process models with data-driven relationships (Schneider et al., 2017; Huntingford et al., 2019; Reichstein et al., 2019; Watson-Parris et al., 2019).  There are a number of different machine learning techniques that have been applied to Earth System studies including: random forests (Rodriguez-Galiano et al., 2012; Yang et al., 2016; McGovern et al., 2017), model tree ensembles (Jung et al., 2010; Yang et al., 2016), empirical orthogonal functions (Yang et al., 2019), principal components analysis (Gosh and Mujumdar, 2008), ANNs (Krasnopolsky et al., 2005; Goyal et al., 2014; Gentine et al., 2018; Nguyen et al., 2018; Buckland et al., 2019; Wu et al., 2019), and more.

We have chosen ANNs over other machine learning methods, because once trained, they are computationally efficient, as most of the computational demand is required during the training phase. Perhaps this is one of the main reasons for its large applicability in several different studies, an advantage over more resource-demanding machine learning techniques. Although ANNs are highly data dependent, sometimes leading to problems of over-fitting and generalizations, ANNs are a good approach for problems with large datasets because they allow any number of input variables.

The main goal of this paper is to investigate, step-by-step, the effects of the atmospheric aerosol on the variability of $CO_2$, sensible (H), and latent (LE) heat fluxes in the Amazon region over the deforestation arch, based on the study of three specific experimental sites. The use of ANNs allows to build up a statistical modeling framework relating meteorological (i.e., temperature, humidity, friction velocity) and radiative variables (i.e., incident photosynthetically active radiation (PAR) at the top of the canopy, the percentage of diffuse PAR) to mass and energy fluxes in the forest in a first approach to address the radiative impacts on mass and energy fluxes. The follow up analysis is based on the derivation of a second group of ANNs relating radiative variables with meteorological ones in order to identify the second order effect of aerosols on surface fluxes. Finally, we make use of the radiative transfer (RT) theory with a process-based RT model to create a hypothetical scenario without aerosols and estimate the potential fluxes of mass and energy, not only considering the first order impact of aerosols, but also the second order impact by making use of two distinct groups of ANNs.

With this approach, this study evaluates: (i) the impacts of aerosol optical depth at 550 nm on fluxes of heat, water, and $CO_2$; (ii) the characteristics of aerosol present in each of the experimental sites; and (iii) the isolated impact of aerosols on mass and energy fluxes through statistical modeling with ANNs, in comparison to other environmental factors, such as temperature, moisture, and turbulence.

## 2 Materials and methods

### 2.1 Sites

The deforestation arch consists of 248 municipalities in Brazil going from the state of Rondônia to the state of Maranhão, across the Brazilian savannah and Amazonian biomes (Araújo Junior et al, 2001), about 3,000 kilometres long and up to 600 kilometres wide. Between July and October 1987, the region of the deforestation arch lost approximately 50,000 km of forests in the states of Pará, Mato Grosso, Rondônia, and Acre, and released more than 500 million tons of carbon into the atmosphere, in which 44 million tons in the form of $CO_2$ (Song et al., 2015). The three flux sites evaluated in this paper are in the Brazilian deforestation arch (**Fig. 1**).

### 2.1.1 Bananal Island

The Bananal Island is the largest island in a river on the planet, with approximately 2 million hectares and almost 360 km in the North-South direction and 80 km in the East-West. The Bananal Island is a floodplain region. The flux tower of Bananal Island was installed in the Northern part of the island, in the limits with the Canton State Park, by the Laboratory of Climate and Biosphere of the Institute of Astronomy, Geophysics, and Atmospheric Sciences of the University of Sao Paulo. The average height of the canopy is approximately between 15 to 18 m. The instrumental platform was mounted in a 40 m high structure located approximately 2 km east of the Araguaia River, at the geographic coordinates 9° 49.27' S; 50° 08.92' W, at 120 m a.s.l. The data monitoring started on the 27th October 2003 and the project was based on the facilities of the Canguçú Research Centre of the Instituto Ecológica - UFT, approximately 20 km south of the micrometeorological tower (Borma et al., 2009).

### 2.1.2 Rebio Jaru

In the north-eastern part of the state of Rondônia, Brazil, the Jaru Biological Reserve has approximately 60,000 ha. The entire basin of the Tarumã river is in the reserve and covers about 75% of the unit. The land has a large number of small streams and springs keeping a moist vegetation throughout the year. The vegetation is mainly characterized as an Open Tropical Forest with some palm trees. There are also few areas with lianas and bamboo trees, as well as small spots of dense ombrophiles forest, characteristic of the Amazon region (IBAMA, 2006). The average height of the canopy is approximately 35m, but some trees can reach up to 45m. The tower is located 1,240 m from the river bank in a more preserved region (Gomes, 2011). Data from the Rebio Jaru were collected from August to October 2007, at the geographic coordinate 10° 04.71' S; 61° 56.01' W 148 m a.s.l. Eddy-covariance instruments were installed in the 62 m height (Bosveld and Beljaars, 2001).

### 2.1.3 Sinop

This study site is located in an area of logging with forest management in Fazenda Maracaí. The farm is located to the northwest, approximately 50 km, from the centre of Sinop, Mato Grosso, Brazil, at coordinates 11° 24.70' S; 55° 19.39'W, at 423 m a.s.l. The Maracaí farm consists of a transitional tropical forest (between Brazilian savannah and open ombrophiles forest) with a continuous canopy, composed of trees from 25 to 28m in height (Vourlitis et al., 2008). The vegetation consists of perennial trees that is characteristic of the transition forest in Mato Grosso. The diversity is high, and there is no predominance of a single tree species. There are approximately 80 species and 35 families of trees with diameter at breast height around 10 cm. The climatological seasonality for the ecotone of the transition forest is similar to that of the tropical and closed forest. The sonic anemometer

was installed at 42 meters height. The site is part of the LBA (Large Scale Biosphere-Atmosphere Experiment in Amazonia) integrated project and has been carrying out measurements of this nature since 1999.

## 2.2 Flux tower data

Surface fluxes of $CO_2$, sensible heat, and latent heat were obtained through the eddy covariance technique applied to data collected with sonic anemometers and infrared gas analyzers installed in flux towers located at Bananal Island, Sinop, and the Jaru Biological Reserve (Rebio Jaru). Fluxes were obtained with temporal resolution of 30 minutes. Measurements of meteorological variables, such as temperature, relative humidity, and wind speed were collected. Except for the tower in Sinop, the other towers presented radiative measurements such as incident solar radiation and, incident and reflected PAR. For Sinop, these variables were calculated by a radiative transfer model, libRadtran, further described in Section 3.1.

Geographic coordinates, altitudes, flux tower heights, and the type of sensors used in each one of the three sites are presented in **Table 1**. A Multifilter Rotating Shadowband Radiometer data provided aerosol optical depth at 550 nm estimates per minute and 30 minutes averaged values were calculated, available only for the Rebio Jaru site.

The study period of this work was limited to the dry season, since **(a)** the maximum numbers of fire outbreaks occur in August, September, and October. **(b)** In terms of the temporal distribution of the aerosol optical depth, there is a significant increase in the months of August, September, and October. And, **(c)** the analysis of the intrinsic properties of the aerosols becomes important because it has indications of its chemical composition and size of the particles. Only the months of August, September, and October were considered in this analysis in order to keep consistency of the type of aerosols being evaluated.

## 2.3 MODIS data

The aerosol optical depth (AOD) at 550 nm used in this study was obtained from the Moderate Resolution Imaging Spectroradiometer (MODIS) for Bananal Island and Sinop. Except for Rebio Jaru, where AOD values were obtained in situ with shadow band radiometer (MFRSR) (**Table 1**). MODIS is an instrument on board of the Terra and Aqua satellites. The Terra's orbit around the globe is synchronized and flies from north to south across the equator line in the morning (~ 10:30 local time), while the Aqua presents passages from south to north by the equator in the afternoon (~ 13:30 local time). MODIS observes the entire ocean surface of the Earth and almost the entire terrestrial surface in a period of 1 to 2 days and acquires data in 36 spectral bands. The aerosol product from MODIS is the aerosol optical depth measured over the oceans and a portion of the continents, globally. In

addition, the aerosol size distribution is derived over the oceans, and the aerosol type over continents. Daily MODIS level-2 data is produced with a spatial resolution of 10 x 10 km$^2$ over nadir. The level 2 product was chosen due to its estimation of the aerosol retrieval uncertainty to assist with error analyses as well as a best estimate of the aerosol optical depth data with quality assurance flags already applied. Although spatial resolution

is courser at level-2, lower product uncertainty improved our analysis.

Two groups of MODIS aerosol data products were collected: MOD04_L2, containing the data collected from the Terra platform, and MYD04_L2, containing the data collected from the Aqua platform. The MODIS aerosol product is used to build the climatological variance of aerosols, sources and sinks of specific types of aerosols (e.g., sulphate aerosols, biomass burning), the interaction of aerosols with clouds, and atmospheric corrections of

surface reflectance obtained from remote sensing.

Pre-assumptions about the general structure of aerosol size distribution are necessary for the inversion of the MODIS data and the distribution is described as two log-normal curves: a unique mode, to characterize the particles of accumulation mode (aerosol radius < 0.5 μm) and a single coarse mode to describe dust and salt particles (aerosol radius > 1.0 μm). The aerosol parameters that are thus recovered are: the relationship between

the two modes (fine and coarse), spectral optical depth, and mean particle size. The quality control of these products is based on the comparison with terrestrial stations and climatology. We used instantaneous and monthly fields of the aerosol optical depth at 550 nm (AOD550nm), referring to collection 051 (Hubanks, 2012), level 2. The pixels over the flux towers were considered. Episodes contaminated by clouds were excluded from the analysis.

The MODIS aerosol data was acquired at the moment of each satellite overpass (Terra and Aqua) over the site of interest, and a single point was used to represent the 30 minutes interval. Uncertainties associated with each variable were estimated from the standard deviations obtained for the 30-minute intervals, except for the MODIS AOD550nm measurements, in which the uncertainty of the AOD550nm in an area of 10 x 10 km$^2$ was estimated following Levy et al. (2009) as,

$\sigma_\tau = \pm(0.05 + 0.15\tau)$                                                                  (1)

In Rebio Jaru, in situ AOD measurement was conducted in a relatively limited period of time, covering only part of the dry season of 2007, yet the high temporal resolution of the aerosol optical depth data (every minute) enabled a more robust analysis of the diurnal cycle. On the other hand, in Bananal Island and in Sinop, the time series of flux measurement was longer (from 2003 to 2008, and from 2005 to 2008, respectively). However, AOD data

was limited to maximum two points a day from MODIS, but the analysis of these experimental sites presented possibilities of characterization of the interannual cycles.

**3 Models**

In this section, we discuss the two models used in this study: a physical process-based atmospheric radiative transfer model, libRadtran (Mayer and Kylling, 2005), and a statistical based model, the artificial neural networks. First, we describe the radiative transfer model; second, we present the machine learning technique, ANNs, and describe the specific methodology used in our experiments. Finally, we describe statistical metrics used to evaluate our results.

**3.1 Radiative Transfer Model**

The radiative transfer model libRadtran1.6-beta (Mayer and Kylling, 2005) was used to perform calculations with the incident solar radiation at the Top of the Atmosphere (TOA) specified for the Amazon region with a tropical atmosphere profile. Ozone concentrations were set to 300 DU based on values obtained by Rosário et al. (2009), who found an average value of integrated ozone content in the atmospheric column of 257 DU. In this paper, surface albedos from the International Geosphere-Biosphere Programme (IGBP) classification (Loveland and Belward, 1997) were used.

For the three sites, the surface type was set to a deciduous broadleaf forest. The day of the year provides information about the Earth-Sun distance, which affects the amount of radiation incident at the TOA, and it was also used as input to the code. The correction for molecular absorption, parameterized from the LOWTRAN model (Pierluissi and Peng, 1985), as well as adopted by the SBDART code (Ricchiazzi et al. 1998) was implemented. The resolution of the corrections for molecular absorption was 0.5 nm at wavelengths below 350 nm and 1.0 nm above. The DISORT method (DIScrete ORdinate Radiative Transfer Solvers) was used to solve the radiative transfer equations because of its versatility and broad use (Mayer et al., 2010). Four streams were used in the simulations performed in this paper. Previous studies have shown that differences in incident irradiance at the surface with at least 4 streams are minimal (Mayer et al., 2011). PAR waveband ranges between 400 and 700 nm and total solar radiation between 280 and 4000 nm.

The input data required to describe the intensive optical properties of the aerosols were obtained from models created with the Mie theory from Rosário (2011), derived from the Aerosol Robotic Network (AERONET) products for three experimental sites: Alta Floresta, Abracos Hill, and Rio Branco. These models carry spectral information about the linear extinction coefficient (Q) in km$^{-1}$, the single scattering albedo ($\omega_0$), and the asymmetry factor for the phase function, calculated from the Henyey-Greenstein method (Henyey and Greenstein,

1941). Spectral resolution of the aerosol model in the PAR spectral region was 25 nm, and over the entire solar spectrum it varied from approximately 10 nm at shorter wavelengths up to 700 nm at longer wavelengths.

Three optical models were developed for the biomass burning aerosol emitted over Amazon, which are different in scattering and absorption properties: a more scattering, an intermediate type, and a more absorbing aerosol type. Also, in this study, sensitivity analysis performed with data from the Biological Reserve of Jaru showed that the more scattering aerosol model is the one that best represented the biomass burning aerosols of that season, associated to higher values of AOD550nm. Therefore, in this study, the aerosol model used for all three study sites was the more scattering type.

The specific input parameters of the radiative transfer equations are: solar zenith angle and the aerosol optical depth at 550 nm. The model provides the integrated value of the downward direct irradiance, the downward diffuse irradiance, and the upward diffuse irradiance, at the altitude determined by the user. All calculations were determined at the surface. From these values, it is possible to obtain the descending global irradiance (the sum of the contributions of the direct and diffuse irradiance incident at the surface) and the diffuse fraction, i.e., the ratio between the diffuse and incident global irradiance at the surface.

### 3.2 Artificial Neural Networks

Artificial Neural Networks (ANNs) have great potential because of their capacity to represent the complexity of the phenomena that are analyzed through them. Traditionally, physical simulation systems have great difficulty in reproducing complex responses of natural events, because unknown processes are often unresolved or represented through simplified approaches called 'parameterization schemes', which can limit forecast predictive skill (Huntingford et al., 2019). ANNs can be considered a network of many relatively simple processors, called units or nodes, in which there is a small amount of local memory. In ANNs, these basic local memory units are commonly layered and communicate through channels or connections, and operate only on their own media, that is, with their local data and their input parameters (Papale and Valentini, 2003). They can be seen as robust parallel processor with natural ability to store knowledge gained from experimentation and make this learning available for future use. This means that, from this method, it is possible to create hypothetical scenarios to evaluate the behavior of the output variables, in relation to possible modifications in the input variables.

ANNs are trained from examples contained in the total set of input data. The "weights" are randomly provided at the beginning of the process and modified according to the rule used in the backpropagation method. In this paper, two methods of resolution of ANNs were used: i) the Multilayer Perceptron (MLP) is a direct and unidirectional power supply model of artificial neural networks, which maps the input data sets to an appropriate set of output

data. The MLP uses the backpropagation technique to train the networks. The most commonly used functions in this method are both sigmoid (Bayram et al., 2016); and ii) the Radial Basis Function (RBF) uses functions of radial basis as a function of activation of the nodes of the neural networks. The RBF method typically has three layers: the input, the intermediate, with a non-linear activation function and the output with a linear function. The input parameters are modeled as a vector composed of real numbers, and the outputs are scalar functions with vector input values (Bayram et al., 2016).

The MLP is a function $f: R^D \rightarrow R^L$, where $D$ is the size of the input vector $x$ and $L$ is the size of the output vector $f(x)$, such that, in matrix notation:

$$f(x) = G(b^{(2)} + W^{(2)}(S(b^{(1)} + W^{(1)}x))) \tag{2}$$

With bias vectors $b^{(1)}$, $b^{(2)}$; weight matrices $W^{(1)}$, $W^{(2)}$, and activation functions $G$ and $s$. $W^{(1)}$ is the weight matrix connecting the input vector to the hidden layer. Choices of $s$ include *tanh* and the logistic sigmoid function with *sigmoid(a) = 1/(1 + e$^{-a}$)*.

The RBF is a function with respect to the origin or a certain point $c$:

$$f(x) = f(\|x - c\|) \tag{3}$$

RBF neural networks are good for novelty detection, because each neuron is centered on a training example, and so inputs far away from all neurons constitute novel patterns; but not so good at extrapolation, because it gives the same weight to every attribute as they are considered equally in the distance computation.

For all ANNs used in this paper, 200 neural networks were generated for each one of the different output variables, from all three study sites, and only the one with the smallest error between modelled and observed output was evaluated. From the total dataset, 70% of the data were randomly chosen and used in the training process of the ANNs, 15% in the validation process, and 15% in the testing process. The set used in the training process has the function of exemplifying relations between inputs and outputs to the ANN. This data partitioning was selected to allow the vast majority (> 2/3) of the data to be used for training, and the remaining data (less than 1/3) were equally partitioned between validation and testing, because both processes are thought to have the same importance on the ANN construction.

We have used the two most recommended back-propagation algorithms for optimization of ANNs (Rojek, 2017), the Broyden-Fletcher-Goldfarb-Shanno (BFGS) and Radial Basis function training algorithm (RBFT), and the sum of squares (SOS) to choose between MPL and RBF by selecting the one which presented the smallest deviation to the training data.

The variables used to train the ANNs were: incident photosynthetically active radiation incident at the top of the canopy (PAR$_i$) in Wm$^{-2}$, the partitioning  of photosynthetically active radiation in its diffuse form (PAR*dif*) in

percentage, derived from the method of Reindl et al. (1990), the solar zenith angle (SZA) in degrees, the average air temperature in the canopy (T) in degrees Celsius, the vapor pressure deficit (VPD) in kPa, the friction velocity ($u^*$) in $ms^{-1}$, and the aerosol optical depth at 550 nm (AOD550nm). Therefore, seven variables were used as inputs of the ANN and the flux of carbon dioxide ($FCO_2$), in $\mu mol\ CO^2\ m^{-2}s^{-1}$, was selected as output.

The best carbon flux ANN for all sites presented similarities such as: all three were generated from the RBF training model, used a Gaussian function for the activation of the hidden units, and the identity function for the activation of the output. However, the ANNs presented different number of hidden units. The network generated for the Bananal Island presented 30 hidden units, the one generated for Sinop used 21, and for Jaru, 19. Basically, the larger the number of hidden units in a neural network, the more robust the model is, i.e., the neural network

has a greater capacity to model more complex relationships. At the end, the number of hidden units provides an indication that the database of the three analyzed sites presented different complexities in the relationship between the variables introduced at the moment of the construction of the networks. However, the functions that best managed to describe these relations were the same and the difference between them is the error magnitude between output data used for training and the model output.

The performance of the ANN results was evaluated using the coefficient of determination ($r^2$), which is the Pearson coefficient (r) squared, RMSE and MAE, defined as:

$$r^2 = \left( \frac{\sum_p (y - \text{mean}(y))(t - \text{mean}(t))}{\sqrt{\sum_p (y - \text{mean}(y))^2 (t - \text{mean}(t))^2}} \right)^2 \tag{4}$$

Where $p$ is the total number of data points used for validation only, excluding training and testing, $y$ are the output values of the generated ANNs, and $t$ are the values derived from the flux tower data ($FCO_2$, H, and LE) or

meteorological variables (T, VPD, and $u^*$).

Root Mean Square Error (RMSE) is used to evaluate differences between the values predicted by a mathematical or physical model (y) and the observed values (t). RMSE is a good measure of the accuracy of a model, but only to compare calculations and observations of the same variable, since it is scale dependent. It is obtained by the following equation:

$$RMSE = \sqrt{\frac{\sum_p (y - t)^2}{p}} \tag{5}$$

Finally, the use of the Mean Absolute Error (MAE) does not take into account if an error was overestimated or underestimated, once it is characterized as the average of the errors committed by a model and can be obtained from the following equation,

$$MAE = \frac{1}{p}\sum_p |y - t| \tag{6}$$

## 4. Results

### 4.1 Evaluation of ANNs for $CO_2$, sensible heat, and latent heat fluxes

In the absence of clouds, aerosols were responsible for changing the amount of solar radiation available at the surface and the partition between direct and diffuse radiation. This, in turn, impacted surface temperature, water vapor, among other changes able to affect photosynthesis.

Flux data was combined with meteorological and satellite data. It is important to emphasize that the flux data have a sample interval of 30 minutes, while the satellite data is a 'snapshot' of the atmosphere when the passage occurs in a very short time interval, which registers a value referred to a single minute, but over a 10 x 10 $km^2$ area.

From the integration of the database it was possible to make an analysis using ANNs. Some precautions were taken in order to avoid misleading analysis of the ANNs. For example, values of $u^*$ smaller or equal to 0.2 $m.s^{-1}$ were excluded from the training, testing, and validation processes to avoid uncertain $CO_2$ flux data collected in episodes with less turbulence, and therefore, a less mixed atmosphere. All data points for all variables above or below three standard deviations were excluded from the input datasets of the ANNs because they were considered to be outliers.

Seven variables were used as inputs of the ANNs: photosynthetically active radiation incident on surface (PARi), in $W.m^{-2}$, the partition of photosynthetically active radiation in diffuse form (PARdif), in %, derived from the method of Reindl (1990), the solar zenith angle (SZA), in degrees, the mean air temperature in the canopy (T) in degrees Celsius, the vapor pressure deficit (VPD) in kPa, the friction velocity (u*), in $ms^{-1}$ and the aerosol optical depth at 550 nm (AOD550nm). The flux of carbon dioxide (FCO_2) (in $\mu mol\ CO_2\ m^{-2}s^{-1}$) was selected as output.

**Figure 2** shows the $FCO_2$ data set used for validation, obtained through the eddy covariance method, on the x-axis (Eddy), and the ones calculated by the generated ANN, on the y-axis, with the respective determination coefficient ($r^2$), RMSE, and MAE, for all three sites.

The Rebio Jaru site presented the highest $r^2$ (0.67) of the three evaluated sites, which represents a high correlation between the measured and modelled $CO_2$ flux data. This site presented continuous AOD550nm data coverage throughout the day. At the other two sites, a single day had a maximum of two AOD550nm data points associated with the passages of the Terra and Aqua satellites. This result indicates that a continuous daytime coverage of AOD generated a more robust ANN influencing $FCO_2$ with more predictability.

**Figure 3** shows box and whisker plots for the diurnal cycle of $FCO_2$ (µmol $CO_2$ m$^{-2}$s$^{-1}$), H (W/m²), and LE (W/m²) for flux tower data obtained through eddy covariance (EC) method and modelled with ANNs (ANN) for 6 days (23$^{rd}$ Aug, 24$^{th}$ Aug, 25$^{th}$ Aug, 29$^{th}$ Aug, 01$^{st}$ Sep, and 10$^{th}$ Sep) over the dry season of 2007 in Rebio Jaru. It is possible to notice that on all evaluated days, the ANN results represent well the measured data. Although for the three fluxes ($FCO_2$, H, and LE), the ANNs are not able to capture the extreme values of the distribution (the 10$^{th}$ and 90$^{th}$ percentiles), the median and both quartiles are well represented. During the evaluated period (August, September, and October 2007), the mean value of $FCO_2$ was -12.2 µmol $CO_2$ m$^{-2}$s$^{-1}$, while the average $FCO_2$ generated by the ANN for the same period was -12.4 µmol $CO_2$ m$^{-2}$s$^{-1}$, a relative difference of 1.9%.

The Bananal Island site presented the lowest values of RMSE (3.20) and MAE (2.43). because of its long temporal series, from October 2003 to December 2008, with 320 input points, which was important to establish a more robust mapping of the intra-seasonal variability. Throughout the evaluated period, the mean observed $FCO_2$ for Bananal Island was -7.0 µmol $CO_2$ m$^{-2}$s$^{-1}$, while the average $FCO_2$ generated by the ANN was -7.1 µmol $CO_2$ m$^{-2}$s$^{-1}$, a relative difference of 1.3%.

The Sinop site presented the largest discrepancies between the model and observed $FCO_2$ with a $r^2$ of 0.37, RMSE of 5.69 µmol $CO_2$ m$^{-2}$s$^{-1}$, and MAE 4.78 µmol $CO_2$ m$^{-2}$s$^{-1}$. A database of intermediate size compared to the other sites (237 points), no PAR measurements were performed at Sinop, but values were generated with the radiative transfer model libRadtran. The errors associated with the outputs of the radiative transfer model were included in the neural network training process. The relative difference between the mean observed and simulated $FCO_2$ was 1.2%.

In order to evaluate the radiative effects of aerosol optical depth on turbulent energy fluxes, the same type of analysis performed for carbon dioxide flux, was done for sensible heat flux (H) and latent heat flux (LE). The evaluated period was also the same, August, September, and October, the dry and burning season in the region. For all three experimental sites the ANNs were made from the same variables used in the networks related to $FCO_2$.

All data points above or below three standard deviations were excluded from the input sets, because they were considered to be outliers and the choice of ANNs.

As for $FCO_2$, Rebio Jaru presented the highest $r^2$ (0.87) for sensible heat flux results. However, the same behavior was not observed when evaluating latent heat flux, which presented a relatively lower $r^2$ (0.58) in comparison. In general, the overall results of LE compared to H presented relatively smaller $r^2$ values. The diurnal cycle of H is more well-behaved in relation to LE as it depends on the diurnal cycle of temperature. Latent heat flux, on the other hand, is greatly influenced by advection of moisture coming from other sources not exclusively related to

stomatal opening by the local vegetation. The presence of springs nearby could explain peaks on local maxima of latent heat flux. This can be seen in **Figure 3** in Supplement Material, which shows the diurnal cycle (for some days of the experiment) of measured H and LE fluxes (black dots) and the respective curves calculated from ANNs (red lines).

During the evaluated period (August, September, and October 2007), the mean H observed was 117.4 $Wm^{-2}$, while the average of H generated by ANN for the same period was 111.7 $Wm^{-2}$, a relative difference of 4.9%. For LE, the mean value of the observed data was 218.1 $Wm^{-2}$ and the average value generated by ANN was 218.7 $Wm^{-2}$, a relative difference of 0.3%. Although the analyzes performed between the validation data and those obtained through ANNs presented better results for H, the relative difference between the mean values of LE - calculated

and observed - was lower.

It is important to emphasize that the analysis performed between the validation data and the modelled data generated by ANNs is a way to determine if the neural network is robust. This does not necessarily mean that a low $r^2$ indicates that a given ANN has low predictability. The evaluation of the relative deviation is a way of measuring the accuracy of an ANN. Although it is possible to note in **Figure 2.b** that the ANN of H could better

approximate the measured values, the relative error associated with ANN of LE was around 4.6% lower.

Bananal Island presented intermediate $r^2$ values (0.42 for H and 0.35 for LE), which indicates an average correlation between the observed and the modelled data. For relative deviations, the values were lower than 3% for H data and lower than 1% for LE.

For Sinop, the analyzes of H presented a $r^2$ of 0.53, considered from medium to high performance of the ANN,

and a relative deviation of 0.5%. For LE, the $r^2$ was 0.27 and the relative error was 1.2%.

**4.2 Modeling CO₂ fluxes with ANNs with constant temperature and moisture**

From the generated ANNs, tests were performed to determine the behavior of $FCO_2$ in different scenarios for all three evaluated sites. In the first test, the values of T, VPD, and u* were fixed in order to isolate aerosol radiative effect on the availability of photosynthetically active radiation at the surface and the partitioning of it in its direct

and diffuse forms. This first test is hypothetical, since it is known that the fixed variables also suffer influences of AOD550nm. In the test, AOD550nm varied from 0.0 to 2.0 and the radiative transfer model libRadtran was used to estimate values of $PAR_i$ and $PAR_{dif}$ values, for three different solar zenith angles (15º, 30º, and 45º). The results related to Bananal Island can be seen in **Figure 4**. The other sites showed similar behavior.

The values of T, VPD, and u* were fixed at the 'optimal point' of $FCO_2$, i.e., for which values of T, VPD, and u*,

$FCO_2$ presented the largest absolute values. For the other sites, arbitrary values were fixed, close to those found

at the Bananal Island. **Table 2** shows the values of T, VPD, and u* fixed for each one of the evaluated sites. **Figure 5** shows the variation of $FCO_2$ in relation to AOD550nm for three different SZAs (15º, 30º, and 45º), in the three evaluated sites. For Bananal Island and Rebio Jaru, the increase in AOD550nm generates an initial decrease in the absolute values of $FCO_2$, for almost all SZAs, except for the solar zenith angle of 30º in Bananal Island, which appears to have no impact on $CO_2$ flux until AOD550nm ~ 0.25. However, from this value on, the $FCO_2$ values show a decreasing trend.

The $FCO_2$ in Bananal Island presents a minimum point with the variation of AOD550nm, while in Rebio Jaru, $FCO_2$ presents a more stable behaviour. In the Bananal Island, after the minimum point of the curve, $FCO_2$ starts to increase with AOD550nm, i.e., there is an optimal range of AOD550nm, which is dependent on the SZA. For SZA = 15°, this interval is associated with AOD550nm between 0.59 and 0.93. For SZA = 30°, the interval is narrower, with AOD550nm ranging from 0.57 to 0.61. And for SZA = 45 °, AOD550nm is between 0.45 and 0.55. For higher values of AOD550nm (above 1.75), the behavior of $FCO_2$ also stabilizes, but at different points, dependent on SZA.

In Rebio Jaru, on the other hand, this behavior associated to an 'optimal range' of AOD550nm influence on the $FCO_2$ curve is not observed. It is possible to identify a value of AOD550nm where the value of $FCO_2$ stabilizes and does not present significant variations anymore. This value of AOD550nm, associated to the stabilization of the curves, also appears correlated with the solar zenith angle. For SZA = 15°, this value of AOD550nm is 0.98, for SZA = 30° is 0.88 and for SZA = 45° is 0.65. When analyzing the results for Sinop, this behavior of decrease in the absolute value of $FCO_2$ with the increase of AOD550nm is not verified. The addition of AOD550nm values from 0.0 to 2.0 generates increases in absolute values, that is, the $CO_2$ changes between the local vegetation and the atmosphere decrease with higher aerosol load. This behavior is observed up to a value of AOD550nm from where $FCO_2$ value stabilizes, depending on the SZA. For SZA = 15°, this value of AOD550nm is 1.85, for SZA = 30° is 1.70, and for SZA = 45° AOD550nm = 1.25.

### 4.3 Modeling temperature, humidity, and wind with ANNs

In the first test, temperature, VPD, and u* were fixed as described in **Table 2**, but these variables are directly related to the available energy at the surface and, therefore, these variables would also be impacted in the presence of aerosols. Artificial neural networks were built from the same database used to make the previous ANNs but varying its input and output values.

The first variable to be evaluated was the temperature. Three variables were used as input parameters in the training processes of the temperature ANN: $PAR_i$ (Wm$^{-2}$), $PAR_{dif}$ (%) and SZA (º). These variables were chosen

as inputs for the neural network of temperature because radiation directly impact surface temperature. The available energy at the surface is proportional to $PAR_i$ and $PAR_{dif}$, linked to the diurnal cycle.

The second evaluated variable was VPD. This variable is related to temperature, which is related to $PAR_i$, $PAR_{dif}$, and SZA. Thus, these 4 variables were used as inputs values in the construction of the VPD ANN.

Finally, u* is also affected by energy available at the surface but in a lower degree once it is directly dependent on wind speed. All other variables (except for the AOD550nm itself, in order to avoid overtraining) were used as input values in the construction of the u* ANN.

**Figure 6** shows data used for the validation, i.e., without interference in the construction of the networks. The values of the x-axis correspond to the observed values, and those of the y-axis, to the values generated by the neural networks. The results for the neural networks of VPD, for all evaluated sites, were the most compatible with the values observed and they presented a $r^2$ greater than 0.8. This is mainly because VPD is closely related to temperature. The explanation of the variance of VPD through the variables used as inputs of the ANNs was mainly governed by temperature. In Bananal Island, temperature explained 50.5% of the VPD variance, in Rebio Jaru and Sinop, 65.6% and 38.1%, respectively. Temperature and friction velocity presented a moderate value of $r^2$ (between 0.2 and 0.5) for all cases. However, in the overall sensitivity analysis, $PAR_i$ was mostly responsible for the explanation of the variance of temperature.

### 4.4 Isolating the radiative impact of aerosols on $CO_2$, sensible heat, and latent heat fluxes with ANN modeling

From the generated ANNs, a study was performed in order to identify what the relative differences would be in $FCO_2$, H, and LE fluxes between real observed conditions and an atmosphere without aerosols, i.e., with AOD550nm = 0.0 for the evaluated cases.

First, the radiative transfer model, libRadtran, was used to obtain $PAR_i$ and $PAR_{dif}$ values for an atmosphere with AOD550nm = 0.0. For the calculations, the real values of SZA and day of the year were used. The results for the Rebio Jaru site can be visualized in **Figure 7**. For the other study sites, the same behavior of the curves is observed. It is possible to identify that the observed values of $PAR_i$ are systematically lower than those obtained by the radiative transfer model, while the opposite is observed for $PAR_{dif}$. Aerosol decreases the incident shortwave global radiation at the surface by absorption and scattering processes, while it increases the percentage of diffuse radiation due to the scattering. From $PAR_i$ and $PAR_{dif}$ calculated for an aerosol-free atmosphere, temperature, VPD, and u* were calculated through the ANNs. The SZA values used as input in the libRadtran were the same as those used in the ANNs.

All variables obtained by both models (libRadtran and ANNs) - except for the values of AOD550nm, which were set at zero, and SZAs, which were kept the same as the original database - were used in calculations of fluxes of $CO_2$, H, and LE. **Table 3** summarizes the results. The mean real values and standard deviations of each of the variables of the initial set of data, the mean model values and the respective standard deviations are shown also for all variables. The relative difference between the means of the actual values and the simulated ones, indicating whether the model presented values larger than the real ones (positive), or smaller (negative).

$PAR_i$ and $PAR_{dif}$ variables agreed over the three sites, when comparing the actual and simulated values. The average of the modelled $PAR_i$ values, without aerosol, was 12.4% higher than the average of the real values for the Bananal Island site, 29.8% higher for the Rebio Jaru, and 26.3% % higher for Sinop. On the other hand, the average of the model values of $PAR_{dif}$ was 35.9% lower than the average of the real values for Bananal Island, 52.0% lower for Rebio Jaru, and 38.0% lower for Sinop.

Temperature also showed agreement regarding the increase of the mean values modelled in relation to the mean real values. In Bananal Island, the increase was 11.3%, in Rebio Jaru was 7.4%, and in Sinop the relative difference was 1.7%. The relative difference between the actual and modelled mean values of VPD and u * did not show agreement for the three experimental sites. However, Bananal Island presented a significant increase in VPD (~50%) for an atmosphere with no aerosols.

As demonstrated by Doughty et al. (2010) for a tropical forest, there is a decrease of approximately 13% in the $FCO_2$ for every 1ºC increase in temperature, when it is above 28ºC. Due to the high correlation between VPD and T, the results of the model without aerosols agree with the authors.

For $FCO_2$, Bananal Island and Rebio Jaru presented higher mean modelled values, i.e., a lower exchange of $CO_2$ with the atmosphere is expected under the effect of episodes without aerosols. Sinop presented the opposite effect. This is mainly due to the differences in structural local characteristics of each site. The response of photosynthetic capacity to an increase in diffuse radiation may vary between different ecosystems and seems to be related to the different properties of canopies (Zhang et al., 2010; Kanniah et al., 2012; Tang and Dubayah, 2017). Niyogi (2004) showed that carbon assimilation grows with an increase in aerosol concentration in the case of forests and cultivated areas and decreases over pastures.

The surrounding areas of the flux tower in Sinop show several regions of pasture, which later became soy or maize plantations. For this reason, the less complex canopy structures in the region (with lower vegetation and more homogeneity in the vertical and horizontal distribution) have opposite responses to that proposed by the 'diffuse fertilization effect' theory. The complexity of the canopies in Bananal Island and Rebio Jaru confirms the effect.

The same type of test was performed in order to evaluate the relative differences in H and LE under conditions of an observed aerosol concentration and a simulated atmosphere without aerosols, with AOD550nm = 0.0. In this test, energy (H and LE) fluxes were obtained using variables calculated by the models (libRadtran and ANNs), except for the values of AOD550nm (0.0) and SZAs, which were obtained through the original data. **Table 3** shows the average real values and the standard deviations of the fluxes, H and LE, for the observed database, the average model values and the relative difference between the means of the real values and the simulated ones.

For all three sites, the values of sensible and latent heat fluxes calculated through the ANNs for a scenario without aerosols were higher than the mean observed values. It was observed that, for all experimental sites, an aerosol-free atmosphere was associated with higher $PAR_i$ and lower $PAR_{dif}$ values, which means that in an aerosol-free atmosphere, there are conditions for more available energy at the surface, which may explain the increase in the mean values of sensible and latent heat fluxes. The higher availability of surface radiative energy proved to be used for the increase in temperature. The analysis of energy fluxes indicates that in the presence of aerosols, there is a total reduction in both heat and water exchange between the surface and the atmosphere over the evaluated sites.

Over all experimental sites, sensible heat flux was shown to be more susceptible to changes in AOD550nm compared to latent heat flux. This fact is greatly observed in Bananal Island and Rebio Jaru, but less in Sinop. A possible explanation for this phenomenon can be obtained by evaluating the absorption spectrum of water. The spectral region most affected by the presence of biomass burning aerosols is the PAR and evaluations were limited to this spectrum only. This spectral region interacts very little with water. Another possible explanation is due to the temperature rise of the canopy itself. The vegetation responds to temperature increase with a stomatal closure, which decreases evapotranspiration. Aerosols are more efficient in affecting the exchange of heat between the atmosphere and the vegetation than the exchange of moisture between the two.

## 5. Conclusions

In this study, we evaluated the impact of AOD550 nm on land fluxes of $CO_2$, sensible heat, and latent heat for three study sites located in the deforestation arch region of the Brazilian Amazon using a combination of ANNs and radiative transfer modeling.

Aerosols were responsible for changing the amount of solar radiation available at the surface and the partition between direct and diffuse radiation. The radiative impact of aerosols caused variations in the temperature and humidity, as well as in $FCO_2$. For the impact of aerosol on the flux of $CO_2$, the modelled values with ANN for

Rebio Jaru indicated a strong dependence with solar zenith angle. Temperature and VPD were important in explaining $FCO_2$ variance for all study sites.

For Bananal Island and Rebio Jaru, the increase of AOD550nm initially generates a decrease in the absolute values of $FCO_2$ to AOD550nm ~ 0.25. However, for higher values of AOD550nm, $FCO_2$ decreased, which indicates higher photosynthetic rates. In Sinop this behavior was not observed.

We developed a group of ANNs relating meteorological and radiative variables for three sites in the Amazon region (**Fig. 8**). For Bananal Island, the behavior of decreasing $FCO_2$ with the increase of AOD550nm remained. For Rebio Jaru, $FCO_2$ decreases with AOD550nm up to approximately AOD550nm = 0.50 and showed the opposite behavior after that. For Sinop, the behavior between both scenarios remained basically the same.

In order to obtain the total radiative impact of aerosols on $FCO_2$, a third test consisted of a study to define the relative differences in $FCO_2$ between conditions in the presence of aerosol and without aerosols (AOD550nm = 0.0) in a combination of two distinct methodologies: ANNs and radiative transfer modeling. The average of the modelled $PAR_i$ values without aerosol was higher than the average of the real values for all sites, while the average of modelled values of $PAR_{dif}$ was lower, as expected. Temperature also increased in the modelled scenario without aerosols in relation to the one containing aerosols. The relative difference between the actual and modeled mean values of VPD and u* did not show agreement for the three experimental sites.

Bananal Island and Rebio Jaru presented higher values of modelled $CO_2$ flux in the scenario without aerosol, while Sinop presented the opposite result. This was due to the canopy structural properties of Sinop and surroundings presenting several crop fields. For this reason, the less complex canopy architecture in the region acted to decrease photosynthetic response of the vegetation to less total incident PAR and more radiation in its diffuse form. We conclude that in the absence of aerosols, $CO_2$ flux from the atmosphere to the surface in Rebio Jaru could be reduced to more than half as its current values in the presence of aerosols. This effect is only observable by taking secondary effects into account.

The same type of evaluation was performed for the fluxes of sensible heat and latent heat using ANNs. When evaluating the variables that best explain the variance of sensible heat flux for Bananal Island, VPD and T showed a strong influence of 27.4% and 23.7%, respectively. This fact indicates that energy fluxes of Bananal Island are influenced by temperature. In the case of Rebio Jaru, $PAR_i$ explained about 17% of the variance of H. For Sinop, there is no favorable variable explaining the variances of H and LE. However, friction velocity (u*) presented the highest index of sensitivity (15%), which indicates that the variance of both energy fluxes was more influenced by turbulence itself.

In the first test, the behavior of the fluxes with AOD550nm showed quite similarities in all three sites. The average behavior of both, H and LE, was decreasing by an increase in aerosol optical depth. The increase in aerosol concentration decreases total incident radiative energy at the surface and, thus generates a systematic decrease in H and LE. In the second test, the variation of other meteorological variables affected the fluxes and, although differences were present, the general behavior of energy fluxes with aerosol optical depth remained. Sensible heat flux and latent heat flux decreased with higher AOD550nm.

In the scenario with AOD550nm = 0.0, the values of sensible and latent heat fluxes were higher than the observed values. This indicates that in an atmosphere without aerosols, more energy is available at the surface. Sensible heat flux was shown to be more susceptible to changes in AOD550nm compared to latent heat flux. This fact is reported in the Bananal Island and Rebio Jaru data sets, but less in Sinop. The radiative spectral region most affected by the presence of the biomass burning aerosols is the PAR, and only the aerosol optical depth at 550 nm was evaluated. The radiation at 550 nm interacts little with water vapor. Also, an increase in canopy temperature may lead to stomatal closure decreasing total evapotranspiration, which can impact latent heat flux negatively.

The two group of ANNs developed in this study, i.e., characterizing the first and second order radiative impact of aerosols on mass and energy fluxes, can be used for flux data gap-filling, as well as modeling and evaluating future climate. More evaluations of the impact of aerosols on energy fluxes throughout the Amazon basin are needed, and machine learning techniques are an efficient and accessible tool to develop these studies.

It is important to highlight some important caveats of the ANN methodology used to drawn our conclusions, including, but not limited to: general extrapolations, e.g., a limited number of sites were used in this study, high dependence on sampling together with other data biases, e.g., the dry season only was particularly used in here, the non-consideration of all confounding factors of the examined problem, e.g., a limited number of input variables (7) were used to build the ANNs throughout these experiments. Yet, using restricted datasets and objective choices of features to study rather than an extensive and generic approach remain valid and important. Moreover, expert intervention associated with best practices are expected to diminish these weaknesses associated with machine learning techniques (Reichstein et al., 2019).

Future efforts must include ways to go around the main caveats of this work, including: i) the addition of multi-source (more field sites), multi-scale (the whole Amazon basin represented at different resolutions), and complex spatio-temporal relations (intra and inter-seasonal variability of flux data, as well as advection of aerosols from different sources); ii) uncertainty estimation should be integrated into models; and iii) using physical-based models to test the statistical ones, e.g., linking radiative transfer and ecophysiology models to evaluate the theorical first order impact of aerosols on light scattering and photosynthesis. Deep learning techniques such as

Recurrent Neural Networks (Schmidhuber, 2015) or Long-Short-Term-Memories (Hochreiter and Schmidhuber, 1997) may be able to contour some of these weaknesses (Reichstein et al., 2019).

Nevertheless, coupling machine learning techniques at local and regional levels with land surface models (Moreira et al., 2017) and Earth System Models (Malavelle et al., 2019) can improve the representation of poorly known

process-based mechanisms or add information about unknown relationships between meteorological variables and surface fluxes.

**Code/Data availability**

Flux and aerosol data used in this study are available in https://doi.org/10.6084/m9.figshare.8239691.v1.

**Author Contribution**

R.K.B. designed the study, conducted the analysis, and wrote the manuscript with help from all the other authors. M.A.Y. coordinated the intensive field campaign at Rebio Jaru in the dry season of 2007 and contributed to the writing of the manuscript. N.M.E.R. developed the aerosols optical models. H.R.R., J.S.N., and A.C.A. coordinated the measurements and shared the flux data for Bananal Island, Sinop, and Rebio Jaru, respectively.

**Competing interests**

The authors declare that they have no conflict of interest.

**Acknowledgments**

This work was partly funded by the Coordenação de Aperfeiçoamento de Pessoal de Nível Superior - Brasil (CAPES) - Finance Code 001, at the University of Sao Paulo, Brazil. M.A.Y. acknowledges FAPESP (grant number 06-56550-5) for funding the intensive campaign at Rebio Jaru. This research was carried out in part at the

Jet Propulsion Laboratory, California Institute of Technology, under a contract with the National Aeronautics and Space Administration. California Institute of Technology. Government sponsorship acknowledged. Copyright 2020. All rights reserved. The authors would like to thank the staff from ICMBio at Rebio Jaru. Dr. Renata Aguiar, Federal University of Rondônia, Campus Ji Paraná, and the LBA local office for collecting and processing the flux data. Also, thanks to Mr. Frederico and students of the LBA local office, as well as to Mrs. Ruth Souza and

other employees for the support during the field campaign. The authors thank three anonymous reviewers whose comments, whose comments improved the manuscript.

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

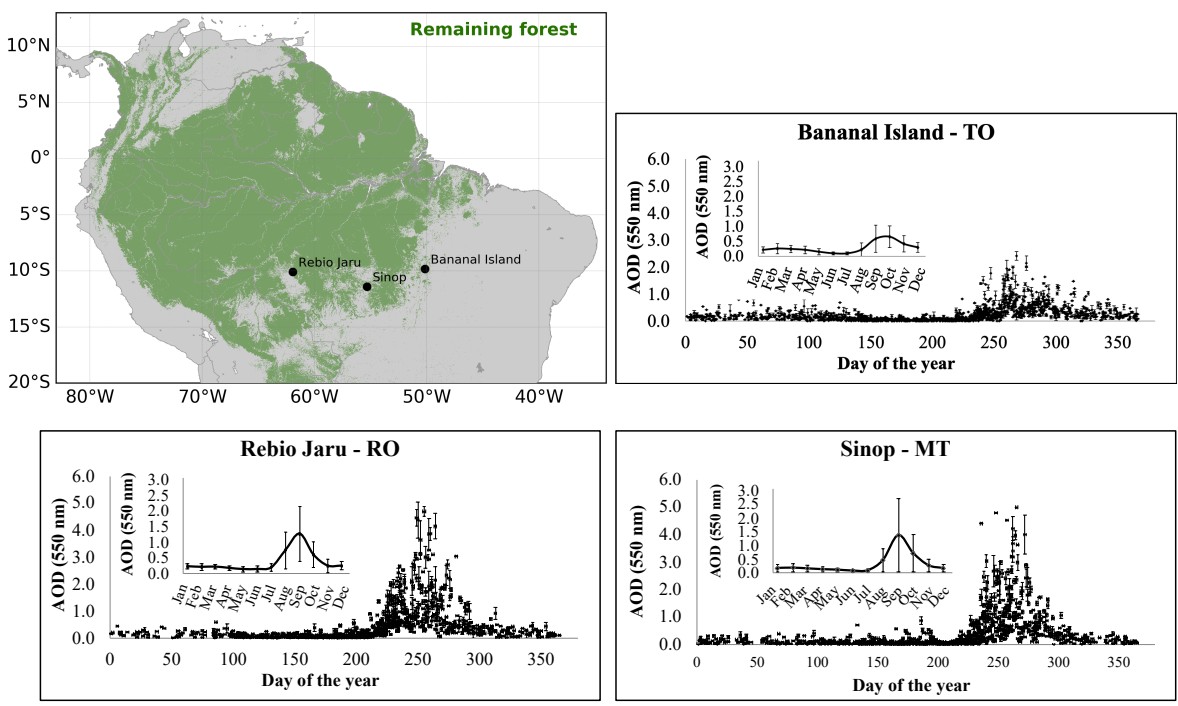

Figure 1: Geographical representation of the North region of Brazil. The darker green area represents the remaining forest. The black dots indicate the location of the study sites: Rebio Jaru, RO (west), Sinop, MT (center), and Bananal Island, TO (east). Boxes show the temporal distribution of the aerosol optical depth at 550 nm over the three sites derived from MODIS from 2003 and 2008 and the monthly mean values for the period, with the standard deviation represented by vertical bars.

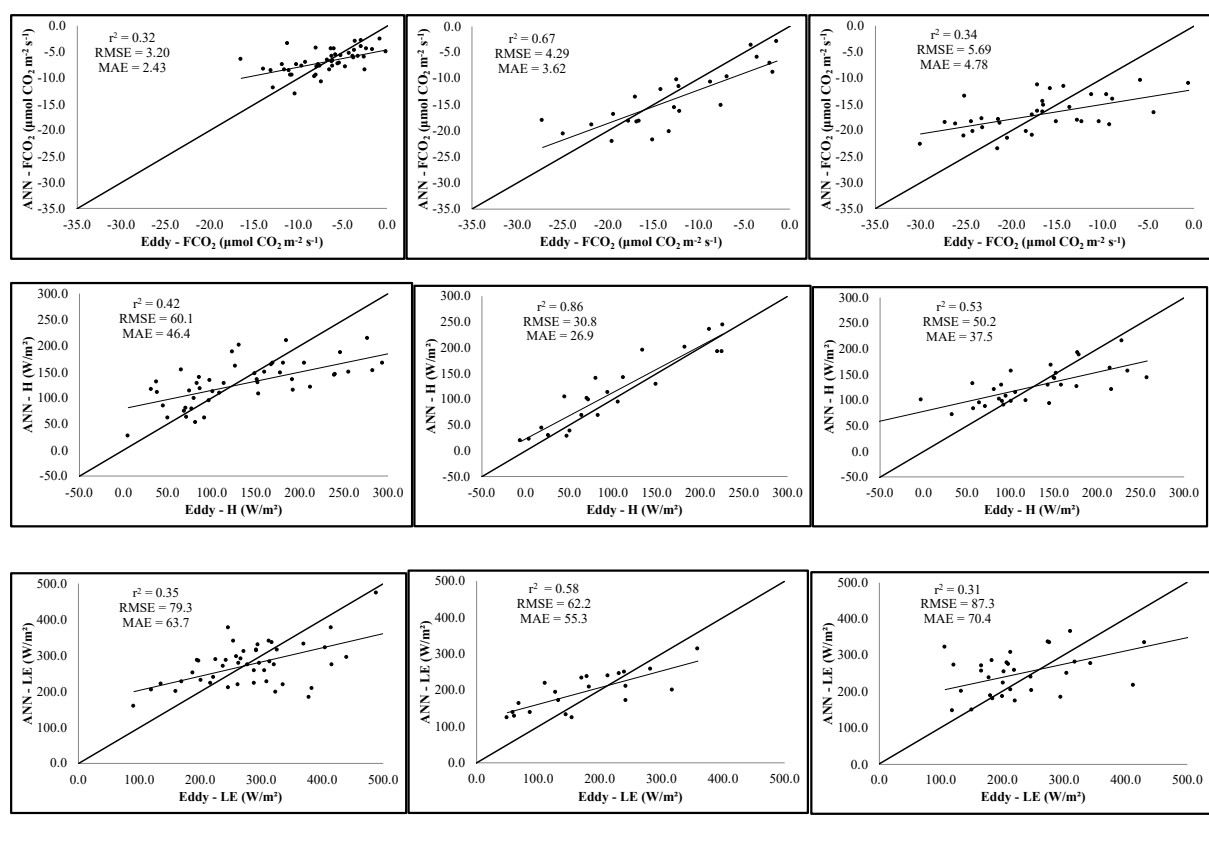

<table>
<tr><td>5</td><td>a. **Bananal Island - TO**</td><td>b. **Rebio Jaru - RO**</td><td>c. **Sinop – MT**</td></tr>
</table>

**Figure 2: FCO$_2$ (µmol CO$_2$ m$^{-2}$s$^{-1}$), H (W/m²), and LE (W/m²) calculated from flux tower data through eddy covariance (x-axis) versus modelled by ANNs (y-axis), with r$^2$, RMSE, and MAE. The 1:1 straight line (thick) and the linear fit (thin) are also represented for the three study sites.**

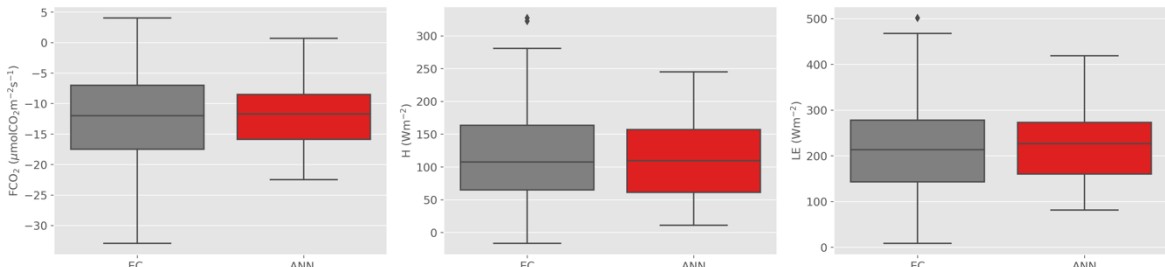

**Figure 3.** Box and whisker plot with extreme data points added for the diurnal cycle of $FCO_2$ (µmol $CO_2$ m$^{-2}$s$^{-1}$), H (W/m²), and LE (W/m²) calculated from flux tower data through eddy covariance (EC) (gray) and modelled by ANNs (ANN) (red) for 6 days (23$^{rd}$ Aug, 24$^{th}$ Aug, 25$^{th}$ Aug, 29$^{th}$ Aug, 01$^{st}$ Sep, and 10$^{th}$ Sep) over the dry season of 2007 in Rebio Jaru. The height of the box portion is given by the interquartile range of the dataset, and extends from the 25$^{th}$ to 75$^{th}$ percentile. The horizontal bar within the box denotes the median value. The ends of the whiskers are drawn to the10$^{th}$ and 90$^{th}$ percentile values. The extreme values are points at the maximum and minimum points.

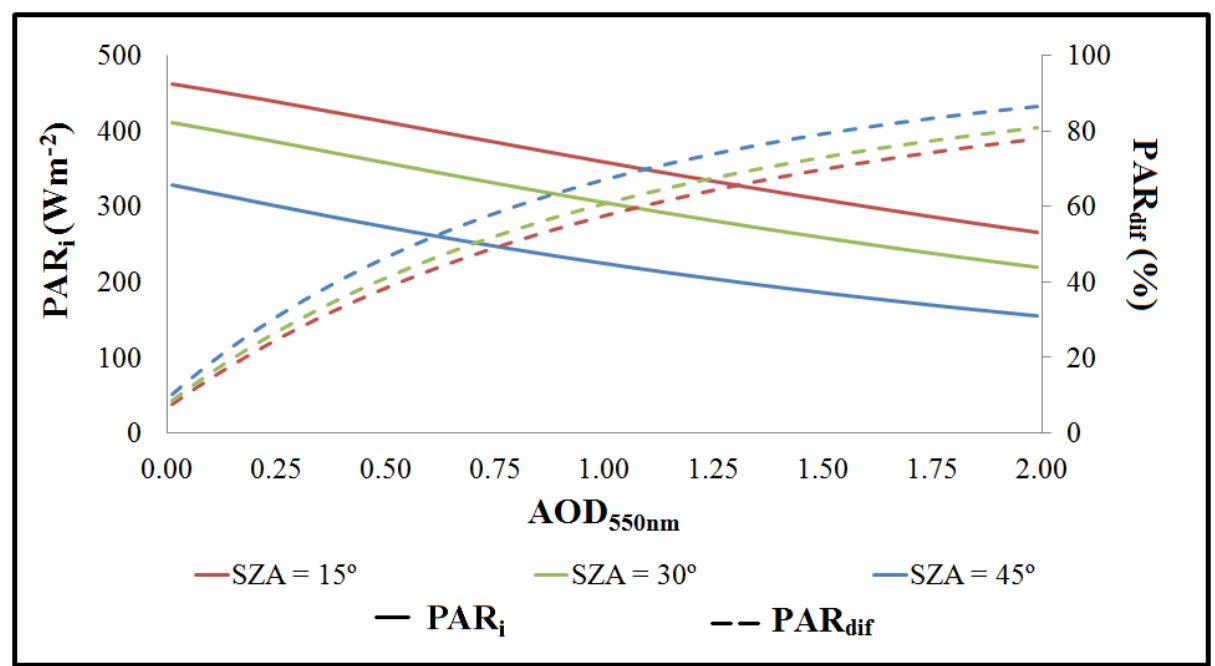

**Figure 4:** PAR$i$ (Wm$^{-2}$) (continuous lines) and PAR*dif* (%) (dashed lines) versus AOD550nm, for three different SZAs (15º, 30º, and 45º) calculated by the radiative transfer model, libRadtran, for Bananal Island for a characteristic day of the dry season.

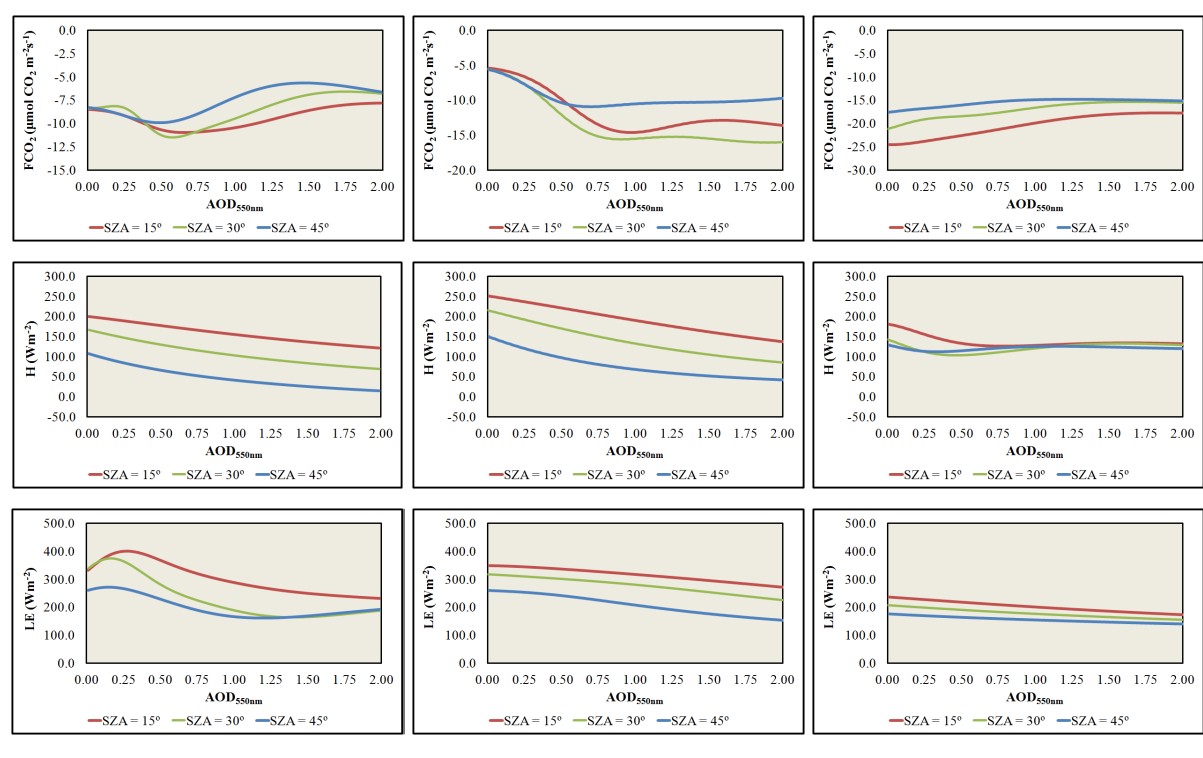

**a.  Bananal Island - TO**     **b. Rebio Jaru - RO**     **c. Sinop - MT**

5    **Figure 5: Modelled FCO$_2$ (μmol CO$_2$ m$^{-2}$s$^{-1}$), H (W/m²), and LE (W/m²) via ANNs versus AOD$_{550nm}$, for three SZAs (15º, 30º, and 45º), for three study sites. Values of temperature, VPD, and u\* were kept constant. PAR$_i$ and PAR$_{dif}$ were obtained through *libRadtran*.**

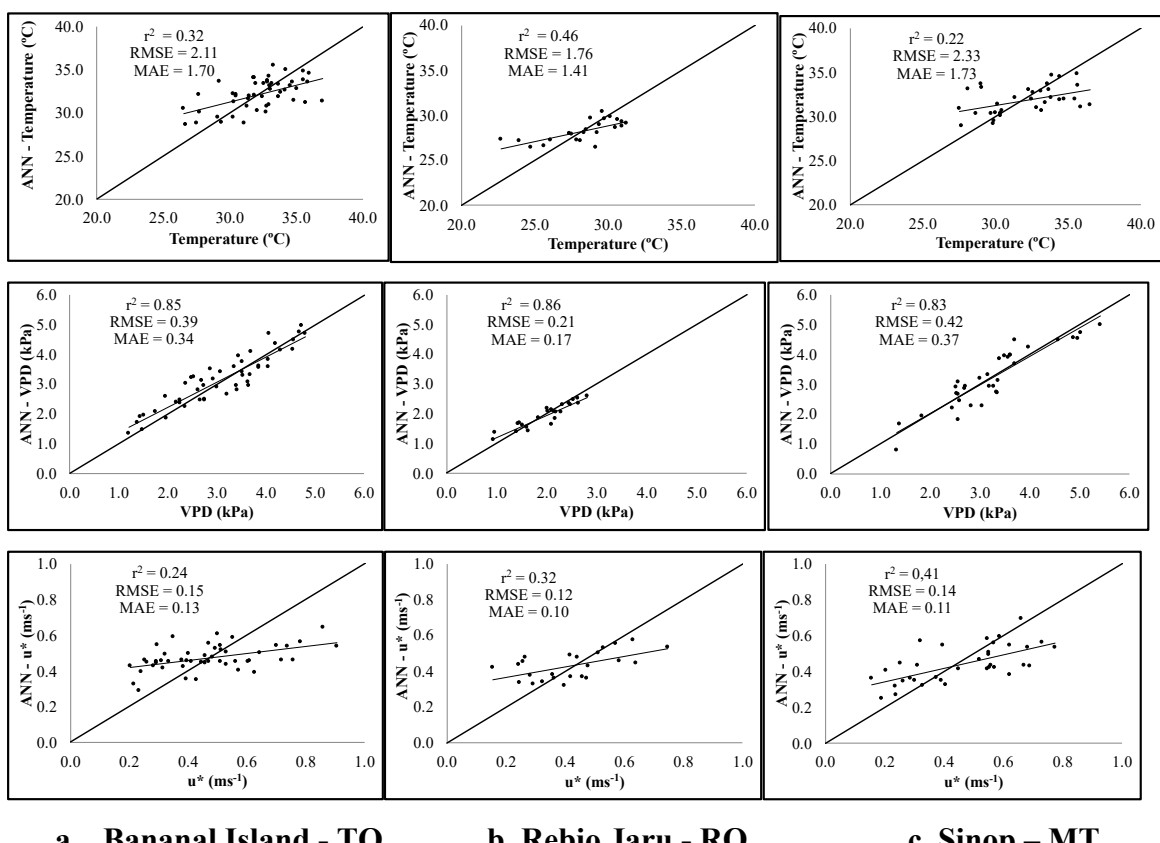

**a. Bananal Island - TO    b. Rebio Jaru - RO    c. Sinop – MT**

5  **Figura 6: Temperature (°C), VPD (kPa), and u\* (m.s⁻¹) obtained from ANNs generated with observed values for the three study sites, with respective r², MAE, and RMSE. The 1: 1 line (thick) and the best linear fit (thin) are also shown.**

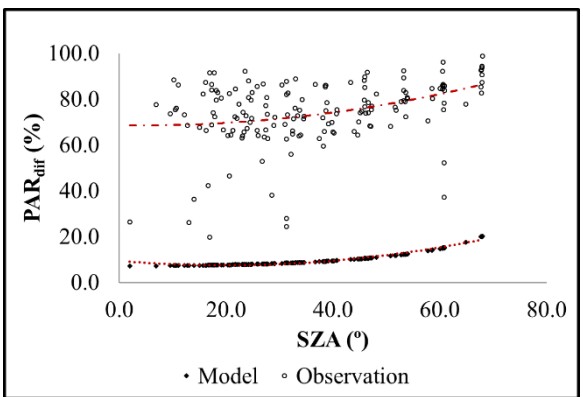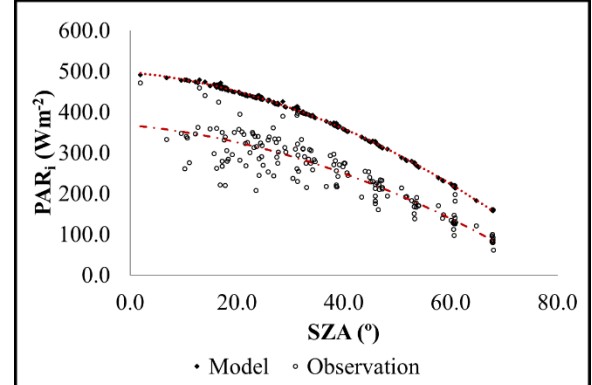

**Figure 7: PAR$_i$ (Wm$^{-2}$) (left) and PAR$_{dif}$ (%) (right) versus SZA (°), for Rebio Jaru. The observed values of PAR$_i$ and PARdif - obtained by the method of Reindl (1990) - are represented by open circles, while the values calculated through libRadtran are represented by the black dots. The curves fitted to the data sets are shown in red.**

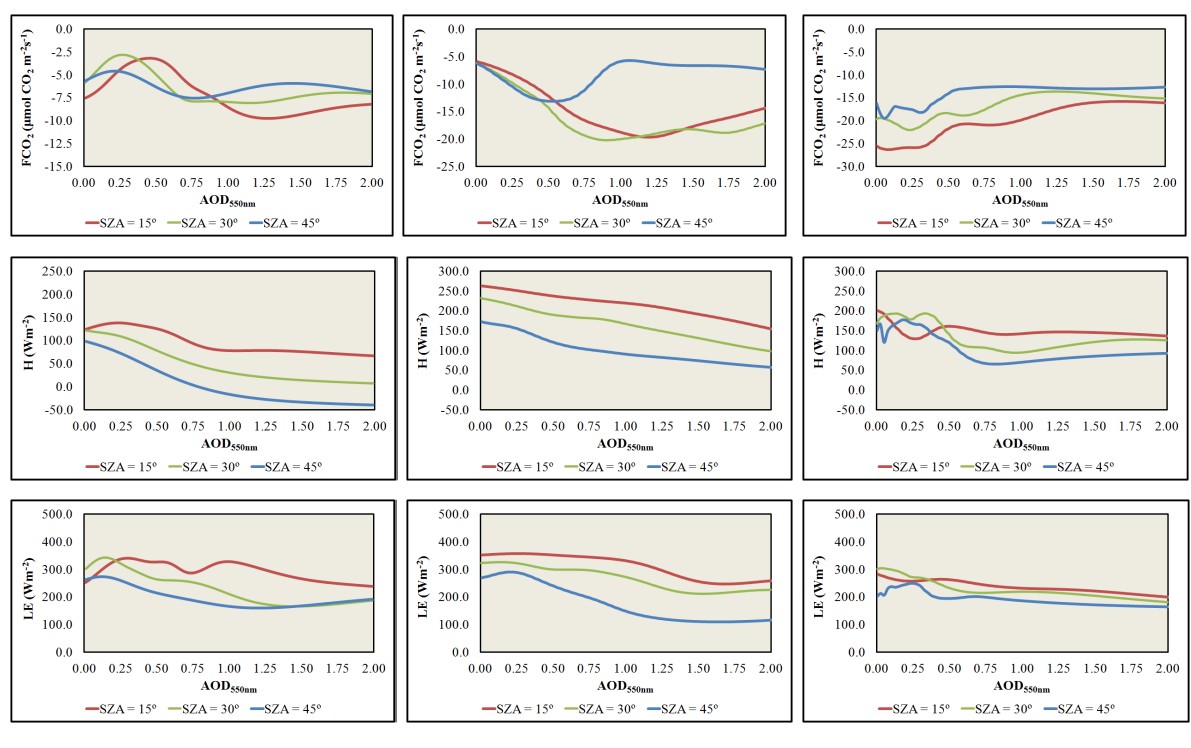

**a. Bananal Island - TO**   **b. Rebio Jaru - RO**   **c. Sinop - MT**

Figure 8: Modelled FCO2 ($\mu mol\ CO_2\ m^{-2}s^{-1}$), H (W/m²), and LE (W/m²) via ANNs versus AOD550nm, for three SZAs (15º, 30º, and 45º), for three study sites. Values of temperature, VPD, and u* were statistically modelled with ANNs built from meteorological data. PARi and PARdif were obtained through libRadtran.

**Table 1: Geographic coordinates, altitudes, flux tower height, and the sensors used in the three experimental sites.**

| | Bananal Island | Rebio Jaru | Sinop |
|---|---|---|---|
| **Coordinates** | 9°49.3'S; 50°08.9'W | 10°04.7'S; 61°56.0'W | 11°24.7'S; 55°19.4'W |
| **Altitude** | 120 m | 148 m | 423 m |
| **Flux tower height** | 40 m | 62 m | 40 m |
| **Period** | Oct/2003 - Dec/2008 | Aug/2007 - Oct/2007 | Mar/2005 - Aug/2008 |
| *datalogger* | CR5000 & CR10X (Campbell Sci.) | CR10X (Campbell Sci.) | CR5000 & CR10X (Campbell Sci.) |
| **Fluxes** | | | |
| **Sonic anemometer** | CSAT3; 10 Hz | Solent 1012R2; 10.4Hz | SWS-211/3K; 10 Hz |
| **Gas analyser** | IRGA, Li-7500 | IRGA, Li-7500 | NOAA-ATDD |
| **Meteorological variables** | | | |
| **Temperature and humidity** | Psychrometer CSI HMP45C | Termohygrometer Vaisala (HMP45D) | HMP-35 Vaisala |
| **Precipitation** | Hydrological Services | Pluviometer EM ARG-100 | 2501, Sierra-Misco |
| **Net radiation** | Net radiometer (Kipp&Zonen) | Net radiometer (Kipp&Zonen) | NR-LITE (Kipp&Zonen) |
| **Radiation** | | | |
| **Solar radiation** | Pyranometer (Kipp&Zonen) | CM-21 (Kipp&Zonen) | *libRadtran*[1] |
| **PAR** | Radiometer (Kipp&Zonen) | Skye SKE-510 | *libRadtran*[1] |
| **Aerosol** | | | |
| **$AOD_{550nm}$** | MODIS | MFRSR[2] | MODIS |

[1] Values obtained from the radiative transfer model
[2] *Multi-Filter Rotating Shadowband Radiometer*

5  **Table 2: Fixed values of temperature (T), vapor pressure deficit (VPD), and friction velocity (u*) for three sites evaluated in the first test.**

| | Bananal Island - TO | Rebio Jaru - RO | Sinop - MT |
|---|---|---|---|
| **T (ºC)** | 28.5 | 29.0 | 29.0 |
| **VPD (kPa)** | 1.5 | 2.0 | 2.0 |
| **u* (m.s$^{-1}$)** | 0.3 | 0.2 | 0.2 |

**Table 3: Mean values with standard deviations of the turbulent fluxes of $CO_2$ ($FCO_2$), sensible heat (H), and latent heat (LE), in µmol $CO_2$ $m^{-2}s^{-1}$ and $Wm^{-2}$, respectively. These values were used as input parameters for building ANNs (Measured), mean values and standard deviations generated from the aerosol free scenarios (Modelled), and relative differences (%) between the measured and the modelled values for all three study sites.**

| Study site | Variables | Measured | Modelled | Relative difference |
|---|---|---|---|---|
| Bananal Island - TO | $FCO_2$ | $-7 \pm 5$ | $-6 \pm 1$ | **+12%** |
| | H ($Wm^{-2}$) | $131 \pm 67$ | $347 \pm 112$ | **+62%** |
| | LE ($Wm^{-2}$) | $287 \pm 108$ | $306 \pm 26$ | **+6%** |
| Rebio Jaru - RO | $FCO_2$ | $-12 \pm 7$ | $-6 \pm 0$ | **+55%** |
| | H ($Wm^{-2}$) | $117 \pm 74$ | $184 \pm 65$ | **+36%** |
| | LE ($Wm^{-2}$) | $215 \pm 98$ | $219 \pm 68$ | **+2%** |
| Sinop - MT | $FCO_2$ | $-18 \pm 7$ | $-20 \pm 3$ | **-12%** |
| | H ($Wm^{-2}$) | $137 \pm 71$ | $162 \pm 24$ | **+15%** |
| | LE ($Wm^{-2}$) | $227 \pm 89$ | $262 \pm 38$ | **+13%** |