# Peer review of "Characterization of the radiative impact of aerosols on CO2 and energy fluxes in the Amazon deforestation arch using Artificial Neural Networks"

_Atmospheric Chemistry and Physics, 2019_

## Referee Comment (RC1) · Anonymous Referee #2 · 24 Jun 2019

Description about the data sets and methods are not provided clearly in the beginning of the manuscript. Reader will only find part of the information in the conclusion section.

ANN technique explained in the "method" section should be focused in relation to the current manuscript.

Some key references are not cited: 1. Aerosols and their influence on radiation partitioning and productivity in northern Australia May 2009, Theoretical and Applied Climatology 100(3):423-438 2. Exploring the link between clouds, radiation, and canopy pro-

ductivity of tropical savannas December 2013, Agricultural Meteorology 182(183):304-313 3. Advantages of diffuse radiation for terrestrial ecosystem productivity, JOURNAL OF GEOPHYSICAL RESEARCH,VOL.107,NO.D6, 4050 4. Fires increase Amazon forest productivity through increases in diffuse radiation, Geophysical Research Letters, 10.1002/2015GL063719 5. Enhancement of crop photosynthesis by diffuse light: quantifying the contributing factors, Ann Bot. 2014 Jul; 114(1): 145–156

The use of MODIS AOD (10 km x 10 km) is not justified well

A very long conclusion. Well, it is not really a conclusion, but rather mixed with the scope of the study, methods and summary of the results. These components should be separated and placed under Introduction, Methods and Results sections respectively.

---

## Referee Comment (RC2) · Anonymous Referee #3 · 5 Nov 2019

**Summary**

This paper presents the evaluation of the relation of aerosol optical depth and surface fluxes of mass and energy on three study sites using Artificial Neural Networks and radiative transfer modeling. The impact of atmospheric aerosols on the carbon balance over three sites in South America were investigated, in the deforestation arch of the Amazon forest, making use of a novel approach, combining two different types of modeling techniques.

**Response overview**

The paper was written well and the field and the problems are interesting. The paper seems novel, by utilising a machine learning approach in this field, but first question, is it the only artificial neural networks (ANNs) or machine learning model available there? I haven't found any discussion or references about this in the manuscript.

The title contains artificial neural networks (ANNs), they need to be defined briefly in the introduction because some ACP readers may not be familiar with this type of model.

In line 14-15 (page 4), it stated; "make use of Artificial Neural Networks (ANNs) to build a new modeling framework from scratch relating a number of state variables", what does it mean by "from scratch" in this context?

There are many other statistics and machine learning methods, the authors must mention clearly and briefly why ANNs were chosen with the pro and cons.

After section 3 Models, please explain briefly what do you want to discuss in the section. Why there are two subsections etc. Perhaps you want to state that there are two types of models used here that are radiative transfer model and ANNs that are physics based and statistical based models.

Since there are two subsections in section 3, Section 3.1 subtitle can be "radiative transfer model." libRadtran is just the name of one of the model.

Lines 22-23 (page 9), state and cite, where it said that: "Traditionally, simulation systems, whether physical or statistical, have great difficulty in reproducing complex responses of natural events." Why are they difficult to reproducing complex responses of natural events? Also in fact ANNs can also be considered as a statistical method, as they learn from data too to construct a model.

Lines 21-30 (page 9), there is a brief explanation about ANNs, then the explanation is continued in the page 10 (lines 1-10). Express the formulation of MLP and RBF

mathematically (even briefly is okay), so the readers get insight about these ANN types without checking other articles or sources and understand the key differences between these two. Also state why you chose them since there are still many other classes in ANNs themselves.

Although, this is not main topic in this paper, but the authors need to state why 70

ANNs usage can also be tricky, in addition to the types of ANN (i.e. the structure), there are many other tuned variables and methods need to be selected, including optimisation method, number of hidden layers, activation functions types, number of neurons in each layer, etc. How did the authors optimise these selection?

Lines 1-13 (page 11), MAE and RMSE are good metrics for modelling. Although Pearson correlation coefficients (PCC) can be used for an additional performance metric, but for modelling, coefficient of determination is typically used, symbolised by $R^2$. It can be obtained simply by taking power two of PCC.

The ANN mathematical expresssions asked to be included in the paper, must match the metrics used here (equations 2-4). For example, line 4 (page 11), explain clearly calculated values and observed values? They are not clear and seem to be picked from a book or somewhere, I mean the definitions are right, but connect them to the models used, etc.

The results of Figure 3 are very good, but it seems too redundant, the subFigures are also not clear because the fonts are too small. Is there any way to compress the Figure results? for instance by making a box plot of these days and compare the real observation and the ANN outputs.

You mentioned some future works in the conclusion, can you elaborate the weaknesses of your method and connect them to your future works please.

---

## Author Response (AR1)

Point-by-point response to the reviews of:

Renato Kerches Braghiere, Marcia Akemi Yamasoe, Nilton Manuel Evora do Rosario, Humberto Ribeiro da Rocha, José de Souza Nogueira, and Alessandro C. de Araújo. Submitted on 07 Jun 2019. **Characterization of the radiative impact of aerosols onCO2and energy fluxes in the Amazon deforestation arch using Artificial Neural Networks**. acp-2019-167

Submitted on 24 June 2019
**Anonymous Referee #2:**

Description about the data sets and methods are not provided clearly in the beginning of the manuscript. Reader will only find part of the information in the conclusion section (**1**)

ANN technique explained in the "method" section should be focused in relation to the current manuscript (**2**)

Some key references are not cited (**3**)
1. Aerosols and their influence on radiation partitioning and productivity in northern Australia May 2009, Theoretical and Applied Climatology 100(3):423-438
2. Exploring the link between clouds, radiation, and canopy productivity of tropical savannas December 2013, Agricultural Meteorology 182(183):304-313
3. Advantages of diffuse radiation for terrestrial ecosystem productivity, JOURNALOF GEOPHYSICAL RESEARCH,VOL.107,NO.D6, 4050
4. Fires increase Amazon forest productivity through increases in diffuse radiation, Geophysical Research Letters, 10.1002/2015GL063719
5. Enhancement of crop photosynthesis by diffuse light: quantifying the contributing factors, Ann Bot. 2014 Jul; 114(1): 145–156

The use of MODIS AOD (10 km x 10 km) is not justified well (**4**)

A very long conclusion. Well, it is not really a conclusion, but rather mixed with the scope of the study, methods and summary of the results. These components should be separated and placed under Introduction, Methods and Results sections respectively. (**5**)

*Authors' response:*

*We thank anonymous referee #2 for evaluating and reviewing our manuscript. We are especially grateful for suggestions to restructure the manuscript in order to make more easily to understand with a more 'natural flow'. We have to the best of our abilities responded to them.*

*We have restructured the Conclusion section according to the comments and suggestions offered by the referee in the specific comments (see details in 1). We have made it shorter and more objective. We have transferred part of the explanation of the datasets earlier in the text to sections 2.2 'Flux tower data' and 2.3 'MODIS data.*

*We have made section 3.2 'Artificial Neural Networks' more concise and focused to the current scientific problem presented in the manuscript (see details in 2).*

*Although we agree that comment 3 might be true, we have tried to add all the main key references in the literature to our knowledge, but we have tried and amended some extra references that came up to us as fundamental in this research field., i.e., Roderick et al. (2001), Oliveira et al. (2007), and Mercado et al. (2009). A good review on the diffuse fertilization effect literature is presented in Kanniah et al. (2012) and we have indicated this paper as a reference. Additionally, more recent and important studies have been referenced in the manuscript, e.g., Fletcher et al. (2018), Persad and Caldeira (2018), Lee et al., (2018), etc. We also thank you for your suggestions of more specific references and look forward to read them and possibly address them in the following phase of the manuscript. (**They now have been added**)*

*Addressing comment 4, we have added extra paragraphs at the beginning and at the end of section 2.3 'MODIS data' in order to better justify the use of MODIS AOD. Basically, MODIS AOD level 2 data (Levy et al., 2015) were used for all analysis in sites without a multifilter rotating band spectroradiometer (MFR) in situ, i.e., Bananal Island and Sinop. For Rebio Jaru, both data sources (MODIS and MFR) were available anda comparison between the two AOD 550 nm products is show in the attached figure from August, 16th to November, 14th 2007.*

*The MODIS level-2 atmospheric aerosol product (MOD04_L2) is provided on a 10x10 pixel scale (10 km at nadir). The level 2 product was chosen due to its estimation of the aerosol retrieval uncertainty to assist with error analyses as well as a best estimate of the aerosol optical thickness data with quality assurance flags already applied. Although spatial resolution is courser at level-2, lower product uncertainty improved our analysis. As for changes suggested in comment 1, we have made the conclusion section shorter and more objective, as well as restructured the manuscript throughout.*

References:

Levy, R., Hsu, C., et al., 2015. MODIS Atmosphere L2 Aerosol Product. NASA MODIS Adaptive Processing System, Goddard Space Flight Center, USA: http://dx.doi.org/10.5067/MODIS/MOD04_L2.006

[Figure]

**Fig. 1**

Received and published on 5 Nov 2019

**Anonymous Referee #3:**

**Summary**

This paper presents the evaluation of the relation of aerosol optical depth and surface fluxes of mass and energy on three study sites using Artificial Neural Networks and radiative transfer modeling. The impact of atmospheric aerosols on the carbon balance over three sites in South America were investigated, in the deforestation arch of the Amazon forest, making use of a novel approach, combining two different types of modeling techniques.

**Response overview**

The paper was written well and the field and the problems are interesting. The paper seems novel, by utilising a machine learning approach in this field, but first question, is it the only artificial neural networks (ANNs) or machine learning model available there? I haven't found any discussion or references about this in the manuscript.

The title contains artificial neural networks (ANNs), they need to be defined briefly in the introduction because some ACP readers may not be familiar with this type of model.

In line 14-15 (page 4), it stated; "make use of Artificial Neural Networks (ANNs) to build a new modeling framework from scratch relating a number of state variables", what does it mean by "from scratch" in this context?
There are many other statistics and machine learning methods, the authors must mention clearly and briefly why ANNs were chosen with the pro and cons.

After section 3 Models, please explain briefly what do you want to discuss in the section. Why there are two subsections etc. Perhaps you want to state that there are two types of models used here that are radiative transfer model and ANNs that are physics based and statistical based models.

Since there are two subsections in section 3, Section 3.1 subtitle can be "radiative transfer model." libRadtran is just the name of one of the model.

Lines 22-23 (page 9), state and cite, where it said that: "Traditionally, simulation systems, whether physical or statistical, have great difficulty in reproducing complex responses of natural events." Why are they difficult to reproducing complex responses of natural events? Also in fact ANNs can also be considered as a statistical method, as they learn from data too to construct a model.

Lines 21-30 (page 9), there is a brief explanation about ANNs, then the explanation is continued in the page 10 (lines 1-10). Express the formulation of MLP and RBF mathematically (even briefly is okay), so the readers get insight about these ANN types without checking other articles or sources and understand the key differences between these two. Also state why you chose them since there are still many other classes in ANNs themselves.

Although, this is not main topic in this paper, but the authors need to state why 70 ANNs usage can also be tricky, in addition to the types of ANN (i.e. the structure), there are many other tuned variables and methods need to be selected, including optimisation method, number of hidden layers, activation functions types, number of neurons in each layer, etc. How did the authors optimise these selection?

Lines 1-13 (page 11), MAE and RMSE are good metrics for modelling. Although Pearson correlation coefficients (PCC) can be used for an additional performance metric, but for modelling, coefficient of determination is typically used, symbolised by R2. It can be obtained simply by taking power two of PCC.

The ANN mathematical expresssions asked to be included in the paper, must match the metrics used here (equations 2-4). For example, line 4 (page 11), explain clearly calculated values and observed values? They are not clear and seem to be picked from a book or somewhere, I mean the definitions are right, but connect them to the models used, etc.

The results of Figure 3 are very good, but it seems too redundant, the subFigures are also not clear because the fonts are too small. Is there any way to compress the Figure results? for instance by making a box plot of these days and compare the real observation and the ANN outputs.

You mentioned some future works in the conclusion, can you elaborate the weaknesses of your method and connect them to your future works please.

*Authors' response (AR):*

*We thank anonymous referee #3 for evaluating and reviewing our manuscript. We are especially grateful for suggestions to clarify a few misleading points in the manuscript in order to make more easily understandable. We also thank your suggestion to re-plot one of our figures into a boxplot, much easier way to visualize our results. We have to the best of our abilities addressed all their comments, as well as responded to the referee's comments (RC) as follows.*

**RC: The paper was written well and the field and the problems are interesting. The paper seems novel, by utilising a machine learning approach in this field, but first question, is it the only artificial neural networks (ANNs) or machine learning model available there? I haven't found any discussion or references about this in the manuscript.**

**The title contains artificial neural networks (ANNs), they need to be defined briefly in the introduction because some ACP readers may not be familiar with this type of model.**

*AR: We have now added a sentence (underline) in the Introduction addressing this point and added a number of references to help the reader.*

*Line 15-16 (page 16): "First, we make use of Artificial Neural Networks (ANNs), a machine learning algorithm that can approximate any nonlinear deterministic function (Gentine et al., 2018), to build a new modelling framework relating…"*

*Line 23-30 (page 4): "Machine learning has significantly advanced in the last decades, being applied to a number of different research areas, and more recently used in Earth System studies as a powerful tool to link physical process models with data-driven relationships (Schneider et al., 2017; Huntingford et al., 2019; Reichstein et al., 2019; Watson-Parris et al., 2019). There are a number of different machine learning techniques that have been to Earth System studies including: random forests (Rodriguez-Galiano et al., 2012; Yang et al., 2016; McGovern et al., 2017), model tree ensembles (Jung et al., 2010; Yang et al., 2016), empirical orthogonal functions (Yang et al., 2019), principal components analysis (Gosh and Mujumdar, 2008), ANNs (Krasnopolsky et al., 2005; Goyal et al., 2014; Gentine et al., 2018; Nguyen et al., 2018; Buckland et al., 2019; Wu et al., 2019), and more."*

**RC: In line 14-15 (page 4), it stated; "make use of Artificial Neural Networks (ANNs) to build a new modeling framework from scratch relating a number of state variables", what does it mean by "from scratch" in this context?**

*AR: We have now removed the term 'from scratch" and re-structured the sentence.*

**RC: There are many other statistics and machine learning methods, the authors must mention clearly and briefly why ANNs were chosen with the pro and cons.**

*AR: We have now added a sentence (underline) in the Introduction addressing this point and added a number of references to help the reader.*
*Line 2-6 (page 5): "We have chosen ANNs over other machine learning methods, because once trained, they are computationally efficient, as most of the computational demand is required during the training phase. Perhaps this is one of the main reasons for its large applicability in several different studies, an advantage over more resource-demanding machine learning techniques. Although ANNs are highly data dependent, sometimes leading to problems of over-fitting and generalizations, ANNs are a good approach for problems with large datasets because they allow any number of input variables."*

**RC: After section 3 Models, please explain briefly what do you want to discuss in the section. Why there are two subsections etc. Perhaps you want to state that there are two types of models used here that are radiative transfer model and ANNs that are physics based and statistical based models.**

*AR: We have now added a sentence (underline) addressing this point.*

*Line 1-7 (page 9): "In this section, we discuss the two models used in this study: a physical process-based atmospheric radiative transfer model, libRadtran (Mayer and Kylling, 2005), and a statistical based model, the artificial neural networks. First, we describe the radiative transfer model; second, we present the machine learning technique, ANNs, and describe the specific methodology used in our experiments. Finally, we describe statistical metrics used to evaluate our results."*

**RC: Since there are two subsections in section 3, Section 3.1 subtitle can be "radiative transfer model." libRadtran is just the name of one of the model.**

*AR: We have now changed that.*

**RC: Lines 22-23 (page 9), state and cite, where it said that: "Traditionally, simulation systems, whether physical or statistical, have great difficulty in reproducing complex responses of natural events." Why are they difficult to reproducing complex responses of natural events? Also in fact ANNs can also be considered as a statistical method, as they learn from data too to construct a model.**

*AR: We have now modified that sentence to:*

*Lines 15-18: "Traditionally, physical simulation systems have great difficulty in reproducing complex responses of natural events, because unknown processes are often unresolved or*

*represented through simplified approaches called 'parameterization schemes', which can limit forecast predictive skill (Huntingford et al., 2019)."*

**RC: Lines 21-30 (page 9), there is a brief explanation about ANNs, then the explanation is continued in the page 10 (lines 1-10). Express the formulation of MLP and RBF mathematically (even briefly is okay), so the readers get insight about these ANN types without checking other articles or sources and understand the key differences between these two. Also state why you chose them since there are still many other classes in ANNs themselves.**

**Although, this is not main topic in this paper, but the authors need to state why 70 ANNs usage can also be tricky, in addition to the types of ANN (i.e. the structure), there are many other tuned variables and methods need to be selected, including optimisation method, number of hidden layers, activation functions types, number of neurons in each layer, etc. How did the authors optimise these selection?**

*AR: We have now added two equations briefly explaining further the construction of ANNs and differences between MPL and RBF. See lines 5-15 (page 11).*

*We also have added a sentence briefly explaining the methodology used to choose between one formulation over the other (line 20-26, page 11):*

*"This data partitioning was selected to allow the vast majority (> 2/3) of the data to be used for training, and the remaining data (less than 1/3) were equally partitioned between validation and testing, because both processes are thought to have the same importance on the ANN construction. We have used the two most recommended back-propagation algorithms for optimization of ANNs (Rojek, 2017), the Broyden-Fletcher-Goldfarb-Shanno (BFGS) and Radial Basis function training algorithm (RBFT), and the sum of squares (SOS) to choose between MPL and RBF by selecting the one which presented the smallest deviation to the training data."*

**AC: Lines 1-13 (page 11), MAE and RMSE are good metrics for modelling. Although Pearson correlation coefficients (PCC) can be used for an additional performance metric, but for modelling, coefficient of determination is typically used, symbolised by R2. It can be obtained simply by taking power two of PCC.**

*AR: We have now squared all Pearson's coefficients throughout the text and figures accordingly.*

**AC: The ANN mathematical expresssions asked to be included in the paper, must match the metrics used here (equations 2-4). For example, line 4 (page 11), explain clearly calculated values and observed values? They are not clear and seem to be picked from a book or somewhere, I mean the definitions are right, but connect them to the models used, etc.**

*AR: We have now amended the sentence and clarified the variables linking models and metrics.*

*Line 18-20 (page 12):* *"Where p is the total number of data points used for validation only, excluding training and testing , y are the output values of the generated ANNs, and t are the values derived from the flux tower data ($FCO_2$, H, and LE) or meteorological variables (T, VPD, and $u^*$)."*

**AC: The results of Figure 3 are very good, but it seems too redundant, the subFigures are also not clear because the fonts are too small. Is there any way to compress the Figure results? for instance by making a box plot of these days and compare the real observation and the ANN outputs.**

*AR: We have now re-plotted these data in a box and whisker plot format. We have also moved the original sub-figures of Figure 3 to Supplement Material and changed the manuscript accordingly.*

*Line 1-7 (page 14):* *"**Figure 3** shows box and whisker plots for the diurnal cycle of $FCO_2$ (µmol $CO_2$ $m^{-2}s^{-1}$), H (W/m²), and LE (W/m²) for flux tower data obtained through eddy covariance (EC) method and modelled with ANNs (ANN) for 6 days ($23^{rd}$ Aug, $24^{th}$ Aug, $25^{th}$ Aug, $29^{th}$ Aug, $01^{st}$ Sep, and $10^{th}$ Sep) over the dry season of 2007 in Rebio Jaru. It is possible to notice that on all evaluated days, the ANN results represent well the measured data. Although for the three fluxes ($FCO_2$, H, and LE), the ANNs are not able to capture the extreme values of the distribution (the $10^{th}$ and $90^{th}$ percentiles), the median and both quartiles are well represented."*

*Line 2-4 (page 15):* *"This can be seen in **Figure 3** in Supplement Material, which shows the diurnal cycle (for some days of the experiment) of measured H and LE fluxes (black dots) and the respective curves calculated from ANNs (red lines)."*

**AC: You mentioned some future works in the conclusion, can you elaborate the weaknesses of your method and connect them to your future works please.**

*AR: We have now elaborated on the weaknesses of our approach and linked them to future work, as required.*

*Line 18 (page 21) – Line 2 (page 22)* *"It is important to highlight some important caveats of the ANN methodology used to drawn our conclusions, including, but not limited to: general extrapolations, e.g., a limited number of sites were used in this study, high dependence on sampling together with other data biases, e.g., the dry season only was particularly used in here, the non-consideration of all confounding factors of the examined problem, e.g., a limited number of input variables (7) were used to build the ANNs throughout these experiments. Yet, using restricted datasets and objective choices of features to study rather than an extensive and generic approach remain valid and important. Moreover, expert intervention associated with best practices are expected to diminish these weaknesses associated with machine learning techniques (Reichstein et al., 2019).*
*Future efforts must include ways to go around the main caveats of this work, including: i) the addition of multi-source (more field sites), multi-scale (the whole Amazon basin represented at different resolutions), and complex spatio-temporal relations (intra and inter-seasonal variability of flux data, as well as advection of aerosols from different sources); ii) uncertainty estimation*

*should be integrated into models; and iii) using physical-based models to test the statistical ones, e.g., linking radiative transfer and ecophysiology models to evaluate the theorical first order impact of aerosols on light scattering and photosynthesis. Deep learning techniques such as recurrent neural networks (RNNs) or long-short-term-memories (LSTMs) may be able to contour some of these weaknesses (Reichstein et al., 2019)."*

List of all relevant changes made in the manuscript:

- Introduction has been expanded, more references have been added, and more information about Artificial Neural Networks have been amended.
- Section 2.3 have been changed to include clearer description on data sets and methods. An explanation about the resolution of 10x10 km$^2$ of MODIS has been added.
- Section 3 has a short introduction paragraph.
- Section 3.1 moved from 'libRadtran' to 'Radiative transfer model'
- Two equations and further explanation on how to build ANNs, as well as the main differences between MPL and RBF. See lines 5-15 (page 11). We also have added a sentence briefly explaining the methodology used to choose between one formulation over the other (line 20-26, page 11):
- All analysis moved from Pearson coefficient (r) to coefficient of determination (r$^2$).
- Figure 3 was made into box-whisker plot.
- Original Figure 3 was moved into Supplement material.
- A discussion on weaknesses of the approach and future work was added into Conclusions.
- Grammar and typos have been now corrected.
- Acknowledgements now contain NASA and Caltech.

[revised manuscript text omitted]